# Impacts of an industrial deep-sea mining trial on macrofaunal biodiversity

Eva C. D. Stewart [1,2] ✉, Helena Wiklund [1,3,4], Lenka Neal[1], Guadalupe Bribiesca-Contreras [1,5], Regan Drennan[1], Corie M. B. Boolukos [1,6], Lucas D. King [1,7], Muriel Rabone[1], Georgina Valls Domedel[5], Amanda Serpell-Stevens[5], Maria B. Arias [1], Thomas G. Dahlgren[3,4,8], Tammy Horton[5] & Adrian G. Glover [1] ✉

In 2022 a large-scale test of a commercial deep-sea mining machine was undertaken on the abyssal plain of the eastern Pacific Ocean at a depth of 4,280 m, recovering over 3,000 t of polymetallic nodules. Here, using a quantitative species-level sediment-dwelling macrofaunal dataset, we investigated spatio-temporal variation in faunal abundance and biodiversity for 2 years before and 2 months after test mining. This allowed for the separation of direct mining impacts from natural background variation, which we found to be significant over the 2-year sampling period. Macrofaunal density decreased by 37% directly within the mining tracks, alongside a 32% reduction in species richness, and significantly increased community multivariate dispersion. While species richness and diversity indices within the tracks were reduced compared with controls, diversity was not impacted when measured by sample-size independent measures of accumulation. We found no evidence for change in faunal abundance in an area affected by sediment plumes from the test mining; however, species dominance relationships were altered in these communities reducing their overall biodiversity. These results provide critical data on the effective design of abyssal baseline and impact surveys and highlight the value of integrated species-level taxonomic work in assessing the risks of biodiversity loss.

Proposals for deep-sea mineral extraction, first discussed in the mid 1960s[1], include the mining of seafloor massive sulfides found at hydrothermal vents[2], mining of cobalt-rich crusts found on deep seamounts[3] and the extraction of polymetallic nodules found on abyssal plains (3,000–6,000 m depth)[4]. Of these, nodule mining currently receives the most attention from the nascent deep-sea mining industry, with efforts focused on the Clarion–Clipperton Zone (CCZ)—a 6 million km² area of the central Pacific Ocean, estimated to hold over 21 billion tonnes of nickel-, cobalt- and copper-rich polymetallic nodules[4].

Extensive baseline biodiversity surveys conducted over the last 50 years in the CCZ have predominantly focused on the metazoan invertebrate fauna living in or on the abyssal sediments[5–9]. The best-studied component of the benthic assemblage is the sediment macrofauna—animals typically ranging in size from 0.3 mm to 2 cm, dominated by annelid worms, peracarid crustaceans and molluscs. Despite low abundance and biomass these macrofaunal communities have relatively high biodiversity, with higher species diversity than found in comparable deep-sea sedimented communities[10], and an estimated

[1]Natural History Museum, London, UK. [2]School of Ocean and Earth Sciences, University of Southampton, Southampton, UK. [3]Department of Marine Sciences, University of Gothenburg, Gothenburg, Sweden. [4]Gothenburg Global Biodiversity Centre, Gothenburg, Sweden. [5]National Oceanography Centre, Southampton, UK. [6]Department of Geography, School of the Human Environment, University College Cork, Cork, Ireland. [7]Centre for Life's Origins and Evolution, University College London, London, UK. [8]NORCE Norwegian Research Centre, Bergen, Norway. ✉e-mail: e.stewart@nhm.ac.uk; a.glover@nhm.ac.uk

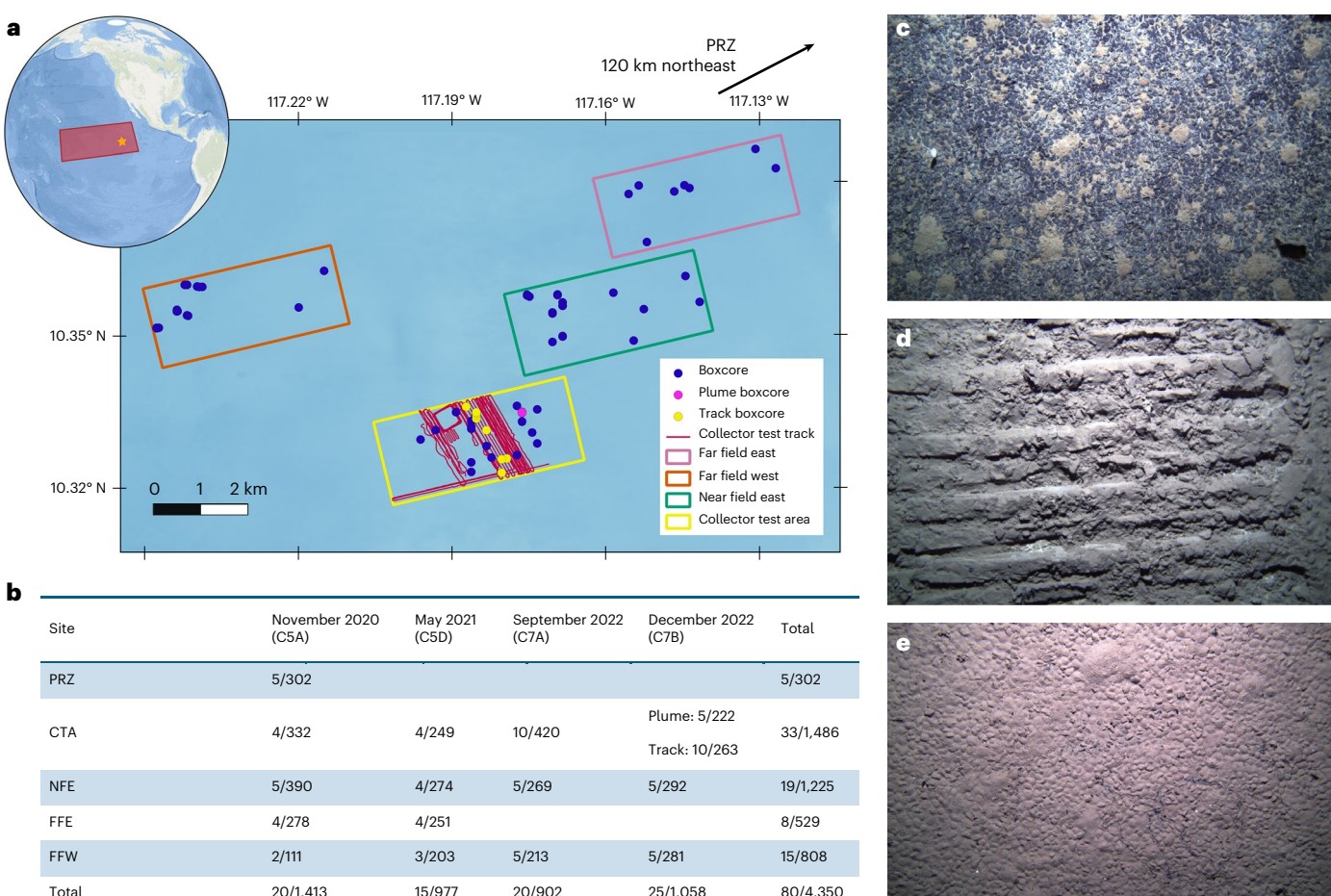

**Fig. 1 | Overview of study region, sampling design and example seafloor morphology. a**, Map of study region sampled in the NORI-D exploration contract area of the CCZ, East Pacific Ocean. Inset map shows location of the CCZ, with the star indicating the location of NORI-D. Points indicate boxcore sample locations taken across all four expeditions, with coloured boxes showing sampled sites. Plume samples were taken in close proximity to each other and so appear as one point on the map. The PRZ sampled in November 2020 is located 120 km to the northeast of the primary sample locations. Full boxcore collection details can be found in Supplementary Table 8. **b**, Overview of samples taken at each location at each time point, numbers show number of boxcore samples/total number of macrofaunal individuals. **c–e**, Example seafloor photographs of each impact level: unimpacted seafloor (**c**); mining tracks (**d**); and plume-impacted area (**e**). Credit: photos in **c–e** by Daniel Jones (National Oceanography Centre).

90% of the collected taxa being undescribed[11]. While the abyssal plains may have once been assumed to host homogenous, static and depauperate communities[12], studies have since shown these habitats to be home to biodiverse and heterogenous communities which can vary markedly across small spatial[13–15] and temporal[16–18] scales.

Tests of nodule-mining vehicles have been carried out in both the Pacific and Indian Ocean since the 1970s, as well as seven published disturbance experiments designed to try and understand the potential environmental effects of seabed mining before commercial-scale operations[19–22]. These studies have produced mixed results: while some report no impact on infaunal (macrofauna and meiofauna) abundance between 2 years and 44 years following disturbance[22–25], others have found significant reductions in macrofaunal abundance immediately following disturbance[26], with residual community-level changes still visible 7 years later[27]. Only one of these studies followed a full-scale contemporary mining test[23], and all have certain methodological limitations such as low statistical replication and lack of pre-impact baseline data.

In October 2022, Nauru Ocean Resources (NORI) a wholly owned subsidiary of The Metals Company (TMC), together with the company AllSeas, conducted an extensive trial of a large-scale prototype polymetallic nodule collection vehicle within the NORI-D contract area of the CCZ (hereafter termed the collector test). This collector tracked over 80 km of abyssal seafloor, within a 2 × 4 km test field, and recovered over 3,000 t of nodules using a riser-pipe—the largest of its kind thus far. Here we report on the results of a large species-level quantitative abyssal sediment-dwelling macrofaunal dataset, collected over 2 years, including the combined analysis of both natural temporal variation and the effects 2 months following a large-scale nodule mining test (Fig. 1). Following a modified Before-After-Control-Impact (BACI) design, we investigated the variability of macrofaunal density and diversity, to distinguish between natural spatiotemporal changes and those directly resulting from mining impacts, providing a critical evidence-base to inform conservation planning and effective management of deep-sea mining.

## Results

A total of 4,350 sediment macrofauna (sensu stricto) were sorted from 80 boxcore samples across four sampling campaigns (from November 2020 to December 2022), from which 3,826 specimens (88%) could be identified to a total of 788 species. The dataset used for statistical analyses included all samples from the Collector Test Area (CTA) – which included two subcategories 'plume' and 'track' taken after impact—and two control sites, Far Field West (FFW) and Near Field East (NFE) (Fig. 1). Samples from the Far Field East (FFE) and Preservation Reference Zone (PRZ) sites were excluded from statistical analyses because of low temporal replication, although they are presented in all

| Site | November 2020 (C5A) | May 2021 (C5D) | September 2022 (C7A) | December 2022 (C7B) | Total |
|---|---|---|---|---|---|
| PRZ | 5/302 | | | | 5/302 |
| CTA | 4/332 | 4/249 | 10/420 | Plume: 5/222 Track: 10/263 | 33/1,486 |
| NFE | 5/390 | 4/274 | 5/269 | 5/292 | 19/1,225 |
| FFE | 4/278 | 4/251 | | | 8/529 |
| FFW | 2/111 | 3/203 | 5/213 | 5/281 | 15/808 |
| Total | 20/1,413 | 15/977 | 20/902 | 25/1,058 | 80/4,350 |

**Table 1 | Observed and average abundance and species diversity per boxcore sample at each site at each time point and for NORI-D as a whole (total)**

| | | nID | Sobs | Percentage of singletons | Percentage unique | Density (ind.m⁻²) | Species richness | Gini-Simpson (1-D) | Simpson's evenness | ES(100)±s.e. | ES(4bc)±s.e. |
|---|---|---|---|---|---|---|---|---|---|---|---|
| November 2020 (C5A) | CTA | 292 | 164 | 64.63 | 8.90 | 332±27.9 | 55.25±2.63 | 0.97±0.008 | 0.633±0.146 | 75.45±3.46 | 164±27.72 |
| | FFW | 98 | 75 | 80.00 | 8.16 | 222±14.1 | 40.5±0.71 | 0.971±0.0002 | 0.861±0.022 | – | – |
| | NFE | 331 | 184 | 65.76 | 11.75 | 313±27 | 49.8±1.92 | 0.974±0.002 | 0.766±0.061 | 76.87±3.51 | 156.4±30.62 |
| | FFE | 248 | 169 | 71.60 | 12.85 | 278±37.9 | 51.5±8.51 | 0.975±0.008 | 0.826±0.095 | 82.79±3.08 | 169±39.32 |
| | PRZ | 261 | 161 | 72.67 | 16.86 | 242±28.2 | 41.8±3.11 | 0.967±0.012 | 0.798±0.212 | 77.29±3.37 | 135.4±22.81 |
| May 2021 (C5D) | CTA | 212 | 139 | 68.35 | 12.26 | 249±48.4 | 45.75±8.38 | 0.974±0.005 | 0.874±0.05 | 79.5±3.05 | 139±26.35 |
| | FFW | 168 | 119 | 78.99 | 7.74 | 271±35.9 | 45±7 | 0.97±0.011 | 0.787±0.146 | 78.29±2.84 | – |
| | NFE | 242 | 147 | 67.35 | 9.50 | 274±22 | 50.25±3.1 | 0.977±0.002 | 0.869±0.022 | 77.44±3.25 | 147±22.6 |
| | FFE | 228 | 141 | 70.92 | 7.46 | 251±33 | 44.5±10.02 | 0.965±0.014 | 0.715±0.16 | 75.07±3.26 | 141±25.92 |
| September 2022 (C7A) | CTA | 369 | 195 | 56.41 | 8.67 | 168±35.7 | 32.5±6.26 | 0.964±0.007 | 0.895±0.051 | 79.52±3.37 | 105.19±20.63 |
| | FFW | 203 | 130 | 69.23 | 10.34 | 170±27.2 | 35±6.36 | 0.967±0.006 | 0.892±0.024 | 76.99±3.09 | 110.6±26.05 |
| | NFE | 221 | 137 | 69.34 | 9.95 | 215±45.9 | 34.8±5.4 | 0.964±0.005 | 0.825±0.07 | 75.91±3.22 | 115.2±23.09 |
| December 2022 (C7B) | FFW | 246 | 157 | 66.88 | 8.94 | 225±46.2 | 42.8±7.73 | 0.973±0.005 | 0.905±0.055 | 80.65±3.13 | 134.4±23.02 |
| | NFE | 262 | 167 | 67.86 | 11.45 | 234±32.8 | 45.2±6.83 | 0.973±0.006 | 0.859±0.079 | 81.63±3.14 | 143.6±30.86 |
| | Plume | 203 | 122 | 65.57 | 7.88 | 178±71.5 | 32.6±10.36 | 0.961±0.011 | 0.844±0.056 | 73.6±3.15 | 103.6±21.56 |
| | Track | 242 | 157 | 70.06 | 8.68 | 106±33.9 | 22.2±7.24 | 0.947±0.018 | 0.946±0.041 | 80.52±3.16 | 77.5±16.1 |
| **Total** | | 3,826 | 788 | 45.25 | – | 217.5±72 | 39.5±11.3 | 0.966±0.013 | 0.845±0.112 | 82.345±3.58 | 129.7±8.5 |

nID, number of individuals identified to a species unit; Sobs, number of species; percentage of singletons, percentage of Sobs that were only found once at that site; percentage unique, percentage of Sobs that are unique to the area; ES(100), expected number of species in a random draw of 100 individuals; ES(4bc), expected number of species in a random draw of four boxcore samples. Values given for density, species richness, Gini-Simpson diversity and Simpson's evenness are mean±s.d. unless otherwise indicated, showing the average density/diversity per 0.25 m² (one boxcore).

figures for visual comparison. The analysed data included 67 boxcore samples, containing 3,519 specimens, from which 3,089 individuals (87.8%) were identified to 692 species. As is typical for abyssal sediment communities, polychaete annelids were the most abundant taxa across the dataset (44.5% of specimens), closely followed by peracarid crustaceans (isopods, tanaids and amphipods; 37.5%) and molluscs (13.7%). Other phyla including Echinodermata, Nemertea, Bryozoa and Cnidaria occurred at much lower abundances, comprising less than 4% of the total faunal abundance. Mean macrofaunal density across all samples was 217.5 ± 72 individuals per m² (ind.m⁻²) (mean ± s.d.) (Table 1).

**Macrofaunal density**

A two-way analysis of variance (ANOVA) found evidence of a statistically significant interaction between site and time on macrofaunal density (ind.m⁻²) when including the track samples in the time series, but not when including the plume (track, $F_{(6,50)} = 8.81$, $P < 0.001$; plume, $F_{(6,45)} = 2.52$, $P = 0.04$) (Fig. 2 and Supplementary Table 1; for discussion of $P$ values see Methods). We found evidence of significant natural temporal declines in macrofaunal density across all sites during the pre-impact period of November 2020 to September 2022 (Fig. 2; see Supplementary Table 2 for all one-way ANOVA results and Supplementary Table 3 for results of pairwise $t$-tests). Similar patterns of change were observed across all phyla (Extended Data Fig. 1). The relationship between macrofaunal density and the multivariate El Niño/Southern Oscillation (ENSO) index (MEI), a potential driver of food availability, is visualized in Extended Data Fig. 2 and indicates a decrease in macrofaunal abundance with an increasingly negative MEI index.

Following the work of Underwood[28,29], we consider an 'impact' to be a difference in the change of mean abundance/diversity, or time course of mean abundance/diversity, in a defined 'impacted' area (in this instance, the mining track or plume) from before to after the disturbance, compared with such changes that occur from before to after in the control locations. Immediately before the collector test

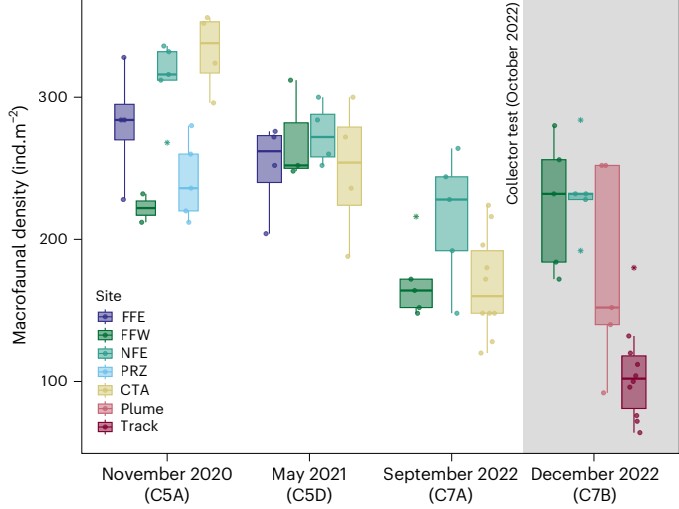

**Fig. 2 | Sediment-dwelling macrofaunal density per m² (median and interquartile range) at each sampled site and time point, before and after collector test-mining impact.** The bars and circles indicate medians and data points, respectively. Outliers are denoted by stars. Boxes indicate the interquartile range (25th–75th percentile) and whiskers indicate lower and upper quartiles. Shaded area highlights samples taken following the collector test. Sample sizes (*n*): C5A FFE = 4, C5A FFW = 2, C5A NFE = 4, C5A PRZ = 5, C5A CTA = 4, C5D FFE = 4, C5D FFW = 3, C5D NFE = 4, C5D CTA = 4, C7A FFW = 5, C7A NFE = 5, C7A CTA = 10, C7B FFW = 5, C7B NFE = 5, C7B plume = 5, C7B track = 10.

(September 2022), there was no evidence for significant spatial differences in macrofaunal density between the control sites and the CTA (ANOVA, $F_{(2,17)} = 3.01$, $P = 0.076$). However, following the collector test, significant differences in density between sites were observed

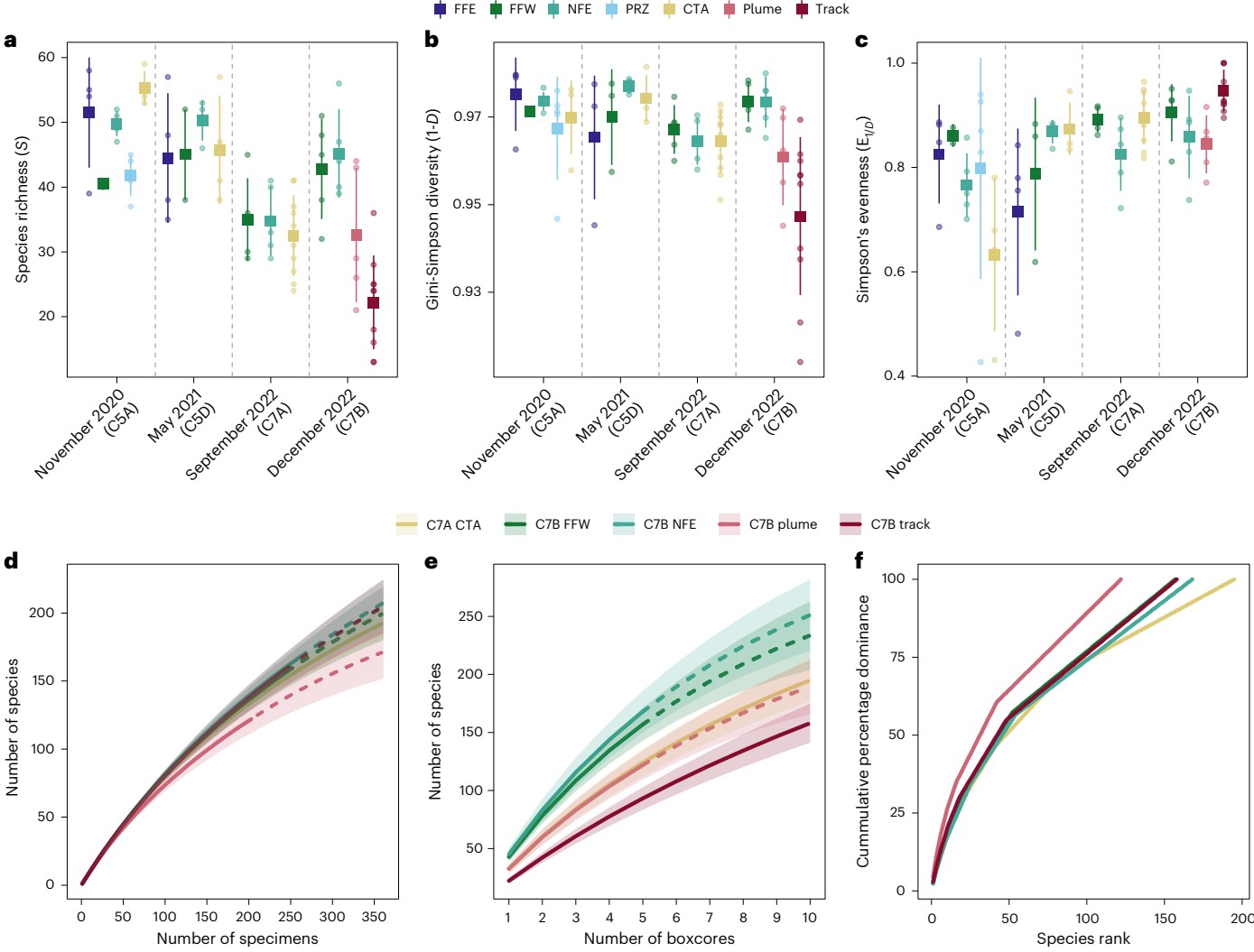

**Fig. 3 | Comparison of sediment-dwelling macrofaunal alpha diversity across sites and time points in the NORI-D area. a–c**, Mean (square) and standard deviation (bar) of different alpha diversity measures at each sampled site and time point, circles show individual data points. **a**, Species richness, measured as number of species per sample (boxcore). **b**, Gini-Simpson diversity index (1-$D$). **c**, Simpson's evenness ($E_{1/D}$). **d,e**, Rarefaction curves of number of species as a function of number of specimens sampled (**d**) and number of boxcores sampled (**e**) in each area. Solid lines are observed values, dashed lines represent

extrapolated values and shading shows 95% confidence intervals. **f**, $k$-dominance curves for each sampled area. For clarity, only sites sampled after test-mining impact in December 2022 (C7B) and the pre-impact collector test area (C7A CTA) are shown. For full curves see Extended Data Fig. 4. Sample sizes ($n$): C5A FFE = 4, C5A FFW = 2, C5A NFE = 4, C5A PRZ = 5, C5A CTA = 4, C5D FFE = 4, C5D FFW = 3, C5D NFE = 4, C5D CTA = 4, C7A FFW = 5, C7A NFE = 5, C7A CTA = 10, C7B FFW = 5, C7B NFE = 5, C7B plume = 5, C7B track = 10.

(ANOVA, $F_{(3,21)}$ = 12.4, $P$ < 0.001). Samples collected from within the mining track had significantly lower macrofaunal densities than NFE ($t$-test, $P$ < 0.001) and FFW ($t$-test, $P$ = 0.002) and non-statistically significant lower densities than those sampled from within the plume area ($t$-test, $P$ = 0.087) (Table 1). Samples taken within the plume area were not statistically different in density than the control sites (Supplementary Table 3).

Macrofauna within the mining tracks were 37% lower in density than 1 month before the collector test ($t$-test, $P$ < 0.001). In comparison, there was no significant change in density observed at NFE ($t$-test, $P$ = 0.489) or at FFW ($t$-test, $P$ = 0.06) across the same period. Densities within the plume were not significantly different from those found in the CTA pre-impact ($t$-test, $P$ = 0.788).

## Species richness and diversity
A total of 788 species were identified across NORI-D. Despite this high number of species, rarefaction curves did not reach asymptotes when rarefied by either specimens or number of boxcores (Extended Data Fig. 3).

Extrapolation analyses estimate needing to sample over 15,000 individuals or over 400 boxcores (equivalent to 100 m²) before the complete diversity of the area is captured. Non-parametric extrapolation analyses predict the total number of macrofaunal species within NORI-D to be between 1,148 and 1,391 (Supplementary Table 4), of which an estimated 521–647 species are polychaete worms.

Spatiotemporal changes in macrofaunal species richness and diversity were examined in the same manner as for macrofaunal density (Methods). There was a statistically significant interaction between site and time on macrofaunal species richness (species per 0.25 m²) when including the track samples, but not when including the plume site (two-way ANOVA, track, $F_{(6,50)}$ = 8.69, $P$ < 0.001; plume, $F_{(6,45)}$ = 2.73, $P$ = 0.02). Results of all two- and one-way ANOVAs can be found in Supplementary Tables 1 and 2.

Concomitant with the observed reduction in macrofaunal density between sampling periods, we found significant natural temporal changes in diversity for both the CTA and NFE sites, but not for FFW. The CTA also showed significant natural temporal changes in evenness,

which were not found in NFE or FFW (Supplementary Table 2). From September 2022 to December 2022, following the collector test, NFE significantly increased in species richness and Gini-Simpson diversity, while there were no significant changes observed in FFW (Fig. 3a–c and Supplementary Table 3). At 1 month before impact, species richness within the CTA was 32.5 ± 6.26 (mean ± s.d.), which significantly decreased within the mining tracks by 32% to a mean of 22.2 ± 7.24 (t-test, P = 0.003) (Fig. 3a). Samples from within the plume site had a mean species richness of 32.6 ± 10.36, which was not significantly different from before impact (t-test, P = 0.985). The same pattern was seen for Gini-Simpson diversity in the CTA (Table 1 and Fig. 3b), with diversity significantly reduced within the tracks, but not significantly different within the plume (Fig. 3b). In comparison, samples taken in the tracks were significantly more even than before impact (t-test, P = 0.026), while evenness per 0.25 m² within the plume was not significantly different from the pre-impact CTA (Fig. 3c).

Increased variability in diversity and composition is often an indicator of disturbance within benthic communities[30] and so Levene's tests were used to examine changes in variance of diversity between sites over time. Gini-Simpson diversity within the mining tracks was significantly more variable than the control sites following mining impact (track-NFE: Levene's test, $F_{(1,13)}$ = 6.45, P = 0.025; track-FFW: Levene's test, $F_{(1,13)}$ = 7.96, P = 0.014) and significantly more variable than the CTA 1 month before impact (Levene's test, $F_{(1,18)}$ = 9.05, P = 0.007) (Supplementary Table 5).

Individual-based rarefaction curves did not reach asymptotes at the local scale for any site at any time (Fig. 3d). For clarity, a reduced number of curves have been presented, including the track and plume, the CTA 1 month before impact (C7A CTA) and the two control sites following the impact (C7B FFW and C7B NFE) (Fig. 3d,e); however, full curves for each site separated by time can be found in Extended Data Fig. 4. When considering individual-based rarefaction—a measure of species accumulation independent of sample size—the communities found within the mining tracks were not notably lower in diversity than other sites. The expected number of species from a random draw of 100 individuals (ES(100)) within the track was 80.5 ± 3.2 (estimated value ± s.e.) compared with a range of 75.1–81.6 from the other sites and time points (Table 1). In comparison, samples taken within the plume area were lower in diversity, with an ES(100) of 73.6 ± 3.2 (Fig. 3d and Table 1). This pattern was also visible when using non-parametric diversity estimators, where the plume has an overall lower number of estimated species than nearly all other sites (Supplementary Table 4).

Alternatively, when looking at sample-based rarefaction curves the track was notably lower in diversity than other sampled sites (Fig. 3e, Table 1 and Extended Data Fig. 3b). The expected number of species from a random draw of four boxcores within the track was 77.5 ± 16.1, compared with an estimated 105.19 ± 20.63 species within the CTA before the collector test (Table 1).

Percentages of species dominance were largely similar between sites and times, with the top ten most-dominant species comprising between 15% and 25.2% of the total abundance at each site (Supplementary Table 6). When plotted on k-dominance curves (Fig. 3f and Extended Data Fig. 4c), the plume is less diverse than other sites, with a greater percentage of common species, a similar pattern as seen in the specimen-based rarefaction (Fig. 3d).

### Community structure and composition

Using permutational multivariate analysis of variance (PERMANOVA) to examine differences in multivariate community composition, we found both site and time to have a significant effect, with no significant interaction between the two, when including both the track and the plume samples in analyses (site: track, pseudo-F = 1.487, P < 0.001 and plume, pseudo-F = 1.375, P < 0.001; time: track, pseudo-F = 1.273, P = 0.002 and plume, pseudo-F = 1.261, P = 0.002) (Supplementary Table 1e). Pairwise PERMANOVA comparisons found significant natural and impact-related

temporal changes in community composition within the CTA, including from 1 month pre-impact to 2 months post-impact within the mining tracks (pseudo-F = 1.364, P = 0.021), while there was no such change observed within the plume (Supplementary Table 3e). No significant temporal changes in community composition were observed for either of the control sites. Following the test-mining impact, community composition within the mining tracks and plume was not significantly different to controls (Supplementary Table 3e).

When visualized using a non-metric multidimensional scaling (nMDS) ordination, there is a large overlap between most sites at each time, with the PRZ as the most distinct from all other sites (Fig. 4a). Samples taken from within the tracks are widely dispersed across the entire plot, overlapping with other sites and times. The multivariate homogeneity of group dispersions was tested using the PERMDISP2 procedure and visualized using a principal coordinate analysis (PCoA) plot (Fig. 4b). We found significant variation in multivariate dispersion between sites (permutation test, permutations = 9,999, $F_{(15,64)}$ = 9.87, P < 0.001), of which, communities within the tracks were significantly more dispersed than all other sites at all time points (Fig. 4c and Supplementary Table 3).

When examining the overall community composition, there is no clear difference in relative proportion of phyla or annelid families between sites and times (Extended Data Fig. 5). Within the mining track, there were lower relative abundances of the families Maldanidae, Nereididae and Sabellidae than there had been in the CTA pre-impact (September 2022) and higher relative abundances of Paraonidae, Spionidae and Phyllodocidae. Indicator species analysis found no significant associations of any taxon to within the mining tracks, either for polychaete family groups or for class-level identifications (Fig. 5 and Supplementary Table 7). Echinoids (sea urchins), however, were significantly associated with the plume (indicator value (IndVal) = 0.571, P = 0.01). For polychaete families, the Orbiniidae were found to be significantly associated with the plume (IndVal = 0.542, P = 0.033).

Only two species were found at all sites at all sampling times: a tanaid crustacean, Stenotanais sp. [NHM_6880_TH_TAN_1] and a benthic serpulid polychaete Serpulidae sp. [NHM_271]. The PRZ, which was only sampled on one occasion and is located 120 km to the northeast of the other sites, had the highest percentage (27.3%) of species unique to the area (Table 1). Of the species found within the track, 13.4% were found nowhere else, while the percentage of unique species at each other sites and time points (excluding the PRZ) ranged from 10.7% to 21.2%. Venn diagrams depicting the number of species overlaps between the CTA pre- and post-impact and the CTA, FFW and NFE sites can be found in Extended Data Fig. 6. The plume site had the highest relative overlap with the track and pre-impact CTA. NFE had more unique species (153) than the CTA (118) and FFW (103) when grouping samples from all time points.

## Discussion

Deep-sea mining within the CCZ is at a critical juncture, as the industry looks to move beyond the exploration phase and into commercial exploitation[31]. Consequently, there is a clear need for direct assessment of the impacts of mining on faunal abundance and biodiversity at the seafloor. Our results show that against a background of strong natural variation we can determine significant impacts of a large-scale deep-sea mining test on several aspects of benthic abundance and diversity. While past studies of smaller disturbances have also been able to show significant effects[21,22,27], this study attempts to disentangle direct mining impacts from the natural variation inherent in abyssal ecosystems via the creation of a temporal baseline.

Immediately following impact, macrofaunal densities decreased significantly by 37% within the path of the nodule collector, while densities within the control sites either increased or remained unchanged. This is in line with patterns seen in previous smaller deep-sea disturbance experiments which have reported 38–63% decreases in

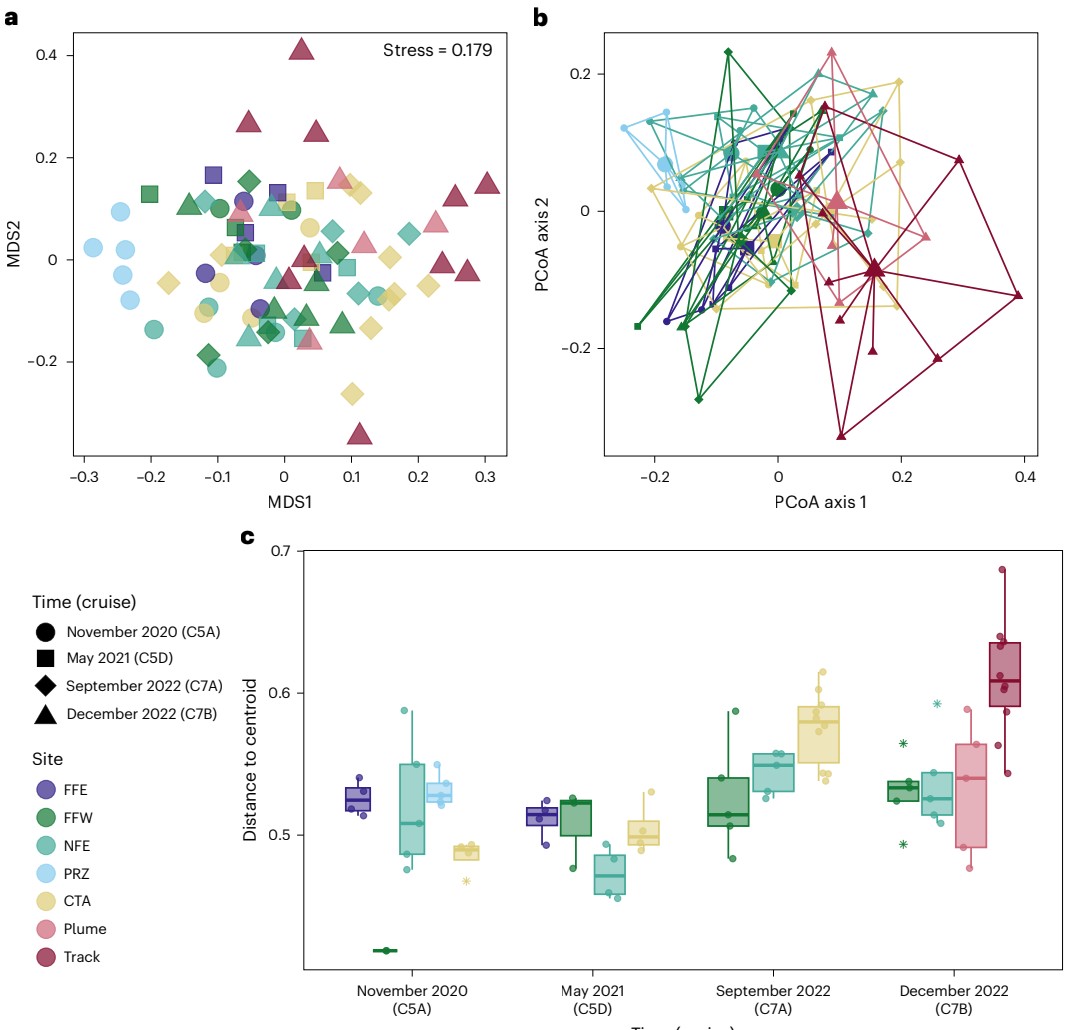

**Fig. 4 | Comparison of sediment-dwelling macrofaunal community composition across sites and time points in the NORI-D area. a**, An nMDS plot of square-root transformed Bray–Curtis dissimilarity between boxcore community data. **b**, Multivariate homogeneity of group dispersions visualized on the first two axis of a PCoA plot. Larger points show group centroids, with smaller points representing individual samples. Individual points for **a** and **b** are coloured by site, with the shape representing sampling period. **c**, Median distance to group centroid of each sampled community within a site. The bars and circles indicate medians and data points, respectively. Outliers are denoted by stars. Boxes indicate the interquartile range (25th–75th percentile) and whiskers indicate lower and upper quartiles. Sample sizes (*n*): C5A FFE = 4, C5A FFW = 2, C5A NFE = 4, C5A PRZ = 5, C5A CTA = 4, C5D FFE = 4, C5D FFW = 3, C5D NFE = 4, C5D CTA = 4, C7A FFW = 5, C7A NFE = 5, C7A CTA = 10, C7B FFW = 5, C7B NFE = 5, C7B plume = 5, C7B track = 10.

macrofaunal density[26,32,33]. Most abyssal infauna occur in the top 2 cm of sediment[8]—within our dataset, 74% of individuals were found in this top layer. As such, the observed reduction in faunal density is unsurprising given that nodule mining machines will directly disturb the top layers of sediment during the nodule-removal procedure.

Sediment plumes generated by nodule mining vehicles have been predicted to cause significant adverse effects on benthic fauna, including damaging gills and feeding structures and increasing mortality due to toxicity from elevated metal concentrations[34]. While some impacts have been demonstrated for megafauna[35], there is limited empirical or in situ data evaluating these effects on macrofauna. Here we found no significant change in macrofaunal abundance or species richness 2 months following sediment plume deposition, as also reported for meiofauna[23] and macrofauna[24] by previous studies. Interestingly, while species richness within the plume area was unchanged, diversity as measured by evenness was reduced across the community as a result of a change in dominance relationships—a similar ecological response to that seen in benthic communities impacted by turbidity flows[36]. Indicator species analysis also found a significant association of the polychaete family Orbiniidae within the plume. Orbiniid species are known

to increase in density in response to pollutants or organic enrichment in shallow waters[37] and the Antarctic benthos[38]. While our sample numbers for individual families are low, it is an interesting observation that certain deposit-feeding taxa may respond to the deposition of mining plumes. This follows results from the DISCOL experiment which found a consistent association of a scavenging polychaete species in the family Sigalionidae with plume-impacted areas[27]. What cannot currently be established is whether this response is caused by the competitive advantage of more resilient species, the mortality of less robust species, or a combination of both. Longer-term sampling and analysis of the impacted site would provide a better assessment of the ecological succession in the community after disturbance.

The impacts of the test mining on faunal density and diversity were observed against the backdrop of significant natural temporal changes. This highlights the importance of collecting temporal data across several control and impact sites to accurately measure the impacts of deep-sea mining. The most likely driver of change in benthic animal abundance is food availability, driven potentially by shifts in the ENSO (MEI)[39]. Macrofaunal densities in NORI-D decreased in parallel with the MEI over the 22-month study period. Significant

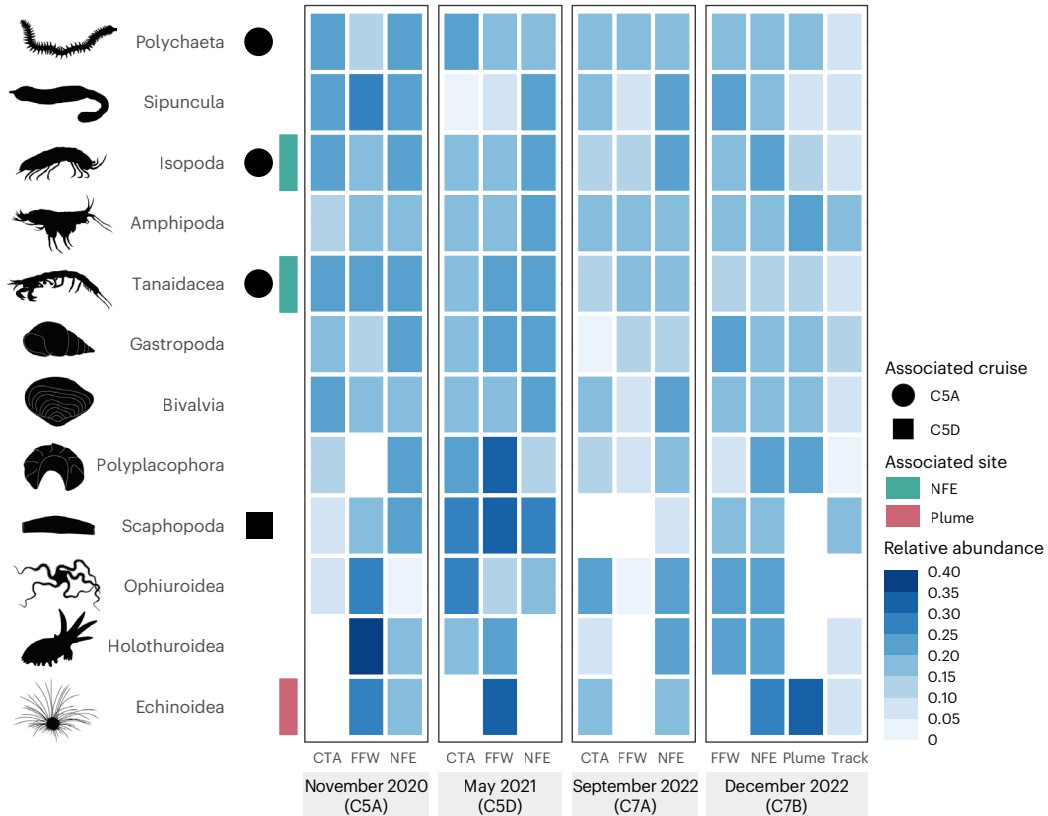

**Fig. 5 | Relative abundance of macrofaunal invertebrate taxonomic groups across sampled sites and times in the NORI-D area.** Cells represent square-root transformed relative abundance of each taxonomic group across all sampled sites. Symbols to the left show significant associations of taxa to specific sites or sampling periods, as identified by Indicator Species Analysis.

seasonal variations in benthic communities have been recorded from across the abyssal Pacific[16,40], with trends extending across interannual periods where food supplies are higher or lower[41]. Without repeated sampling at shorter intervals over longer timescales it will remain unclear as to whether the observed variations in NORI-D are part of stochastic or seasonal variation, or a progressive trend related to longer-term climate change[42]. These data will be crucial to the understanding of long-term recovery rates in disturbed sediments, as significant climate-related changes may have cumulative effects on already impacted communities[43].

Parallel with the observed reduction in faunal abundance, we found a significant decrease in species richness and diversity within the mining tracks. The observed number of species within a sample is intrinsically linked and sensitive to underlying differences in abundance[44]. Therefore, the significantly lower abundance of macrofauna within the mining tracks is the most likely driver of the reduced diversity per sample unit. While previous studies have recorded recovery of macrofaunal abundance after disturbance[22,27], none has published species-level data for the full community. Therefore, there is currently very limited data on whether diversity can or will completely recover following nodule mining. Seven years after the DISCOL disturbance experiment (involving a plough-harrow which did not remove any nodules) the diversity of polychaetes remained reduced with lower evenness and less singleton species[27]. High diversity in abyssal sediments is thought to be maintained through a variety of processes including niche diversification over periods of long-term stability and patch-mosaic dynamics[45]. Given that physical disturbance within the CCZ has been shown to persist over decades[22] and is predicted to last for centuries, it is reasonable to hypothesize that the removal of nodules and associated sediment disturbance will result in persistent long-term decreases in species diversity in the mined areas,

as has been demonstrated to occur in benthic habitats impacted by bottom-fishing[46].

Contrary to this local reduction in species diversity per unit area, diversity measured using sample-size independent accumulators was not significantly reduced within the collector test tracks. All rarefaction curves, at both local and regional scales, indicate that NORI-D remains under-sampled for diversity—extrapolation analyses estimate needing to sample over 15,000 individuals or 400 boxcores (equivalent to 100 m²) before curves begin to reach asymptotes. This hinders the conclusions that can be drawn about differences in diversity. Rarefaction assumes that the area being sampled represents a single habitat[44], whereas the abyssal seafloor is known to be both physically and biologically heterogenous[12,14,47]. As more microhabitats within an area are sampled, the estimated number of species found per number of individuals rises, further exacerbating the 'under-sampling' effect. It is therefore difficult to ascertain if biodiversity within the track is truly in line with unimpacted sites. Reported changes in species diversity following disturbance have been shown to be scale-dependent, with declines more prominent when examined across larger areas[48]. Monitoring changes in diversity patterns from commercial-scale mining operations will require a specifically designed survey to account for this effect. We suggest that such a sampling design should aim to cover representative sites of differing environmental variables such as nodule type and density, both of which have been shown to influence the community composition of different faunal size classes[8,49]. This should occur at sufficient replication both temporally and spatially to encompass natural premining variation, an aspect that has been highlighted as often lacking in deep-sea mining environmental impact assessments[50]. Here we were able to statistically detect variation between sites and time points with two to ten boxcore samples per site/time; however, to improve the robustness of analyses we would recommend a minimum

of five, and ideally ten, cores per site and time with samples spatially randomized within strata[51].

Following the collector test we found significantly higher variability in both diversity and multivariate community composition within the mining track. Increased ecological variability is common following disturbance and is caused by the destabilization of community dynamics, as loss in abundance across populations and the redistribution of resources leads to altered competitive interactions and spatial aggregation of species[52]. The observed even reduction in abundance across broad taxonomic groups, and similar dominance patterns and percentage of singleton species between the track and controls suggest that the collector test caused an equal removal of individuals across the top sediment layers, rather than disproportionately impacting specific taxonomic groups. Such species-specific effects may only be detectable at longer timescales following impact.

While dependent on several ecological and environmental parameters, recolonization and recruitment is a primary determinant of the extent and persistence of disturbance impacts[52]. Knowledge of this is currently very poor in both the CCZ and the broader abyssal ecosystem[53]. Increased spatial variability is exacerbated when species are patchily distributed and when recolonization primarily occurs through lateral migration of adults and juveniles rather than larval settlement, both of which are thought to be the case for abyssal infauna[14,27]. We can therefore hypothesize that in a commercial-scale nodule mine covering 8,500 km² of seafloor[54], community heterogeneity may persist over long timescales. This has already been demonstrated in other deep-sea settings, with elevated variance-mean ratios still observed in the DISCOL area 7 years after impact[27]. Disturbed deep-sea sediments experience highly localized colonization patterns, creating a spatiotemporal mosaic of patches in various stages of succession[55]. Longer-term monitoring of the impacted sites would enable better evidence on rates of recovery. This is particularly relevant for the areas impacted by plume settlement, where ecological changes may occur more slowly than in directly impacted sites and so should be a focus of future environmental impact studies.

In conclusion, through a replicated spatio-temporal sampling programme we detected significant changes in sediment-dwelling macrofaunal density, diversity and community composition in response to a large-scale test of a polymetallic nodule mining machine. This sets a standard for baseline ecological data in the CCZ and further highlights the importance of integrated species-level taxonomic work[56] in assessing ecological responses to disturbance and the risks of biodiversity loss.

## Methods

### Study area and experimental design
The study was conducted in the exploration contract area NORI-D, licenced by the International Seabed Authority (ISA) to Nauru Ocean Resources Inc. (NORI). The NORI-D contract area is located in the southeastern CCZ, occupying an area of approximately 27,000 km² and with water depths of 2,959–4,602 m (Fig. 1). In October 2022, NORI, a wholly owned subsidiary of TMC together with the company AllSeas, conducted a trial of a prototype polymetallic nodule collection vehicle within the NORI-D contract area of the CCZ. This collector test tracked over approximately 80 km of abyssal seafloor within a 2 × 4 km test field and removed over 3,000 t of nodules using a riser-pipe.

The full extent of environmental impacts following a disturbance event can only be accurately detected and assessed when sufficient data are available to disentangle the effect of interest from natural spatial and temporal variability. Considering this, we used an asymmetrical Before-After-Control-Impact (BACI) style experimental design[28,29] with several control sites, following the principles outlined by Etter[51] that formed the basis of the first set of ISA exploration regulations[57].

A random stratified sampling design was used, with data collected from four control sites (FFE, FFW, NFE and PRZ) chosen to be spatially

equivalent to the area being impacted by the mining disturbance and one impacted site, referred to as the CTA. Sites varied in depth from 4,121 m to 4312 m. Samples were collected at three time points before the collector test (November 2020, May 2021 and September 2022) and again 2 months following the collector test (December 2022). Following the impact, samples taken in the CTA were split into 'track' and 'plume' categories. The track samples were collected directly within the mining track. The plume samples were taken 400 m from the mining track. Currently, the sedimentation depth from the impact is not yet published either from modelled or empirical studies[58]. Examination of autonomous underwater vehicle footage taken in the region by the National Oceanography Centre UK team has provided an image of the seafloor in the region of our plume site. This shows a clear veneer of sediment covering the seafloor and the nodules (Fig. 1e). While we do not know the exact depth of the sedimentation, we can confidently state that the plume site 400 m from the track has been impacted by a plume and thus report the impacts on the macrofauna at that distance from the mining track. This will be useful information going forward as further studies are published. A summary of samples taken can be found in Fig. 1, with full details in Supplementary Table 8.

### Sampling
Samples were collected across four expeditions to the NORI-D area spanning a total period of 2 years. Campaign 5A (C5A) took place from October to November 2020 aboard the *Maersk Launcher* and campaign 5D (C5D) took place from April to June 2021 also aboard the *Maersk Launcher*. Campaigns 7A (C7A) and 7B (C7B) both took place on the MV *Island Pride*, from August to September 2022 and November to December 2022, respectively.

Macrofaunal samples were taken with a USNEL Spade Box Core (Ocean Instruments BX-650 and BX-750) with a sample dimension of 50 × 50 cm (BX-650) or 75 × 75 cm (BX-750). Samples taken with the BX-750 were subsampled using a stainless-steel insert frame of 0.25 m² (the same equivalent area as the BX-650, the standard size used on previous CCZ surveys listed above). To ensure boxcores were taken inside the caterpillar tracks created by the collector test, the 'track' cores were visually guided using a remote operated vehicle. The full deployment protocol and subsequent processing of samples followed ref. 59.

In short, following recovery and quality assessment, surface water above the sediment was removed with a hose and passed through a 300 µm sieve. Biota visible to the eye on the sediment surface were removed and fixed separately. All nodules were removed from the sediment surface and washed with cold-filtered seawater (CFSW) over buckets to retain the sediment, which was then included with the 0–2-cm live-sorted layer. The sessile fauna directly attached to the nodules were treated separately and will be the subject of a separate paper. A 15 × 15 cm subcore of sediment was taken and sliced in two layers, 0–2 cm and 2–5 cm, before being transferred to CFSW, sieved through a 300 µm sieve and then taken to the at-sea laboratory where individual animals were picked, identified and photographed from the sediment and preserved individually in 80% non-denatured ethanol for high-quality DNA and morphology. All remaining sediment was sliced in layers of 0–2 cm, 2–5 cm and 5–10 cm and sieved on 300 µm sieves in CFSW, before being bulk fixed in 100% non-denatured ethanol. Sediment residues from the 15 × 15 cm live-sorted fraction were returned to respective bulk-fixed depth layers following the removal and processing of animals. The specimens and data from the live-sorted sample are also subsequently merged into the entire boxcore for analysis.

### Laboratory processing
A total of 80 boxcores were fully quantitatively assessed. All samples were initially sent to the Natural History Museum, London (NHMUK), where the bulk-fixed sediment residues were sorted under stereomicroscopes. Arthropods and annelids were only counted if the head was present, and echinoderms only if the oral disc was present. Sessile

animals such as bryozoans (excluding Ctenostomatida, which are free-living within the sediment), brachiopods, crinoids and sponges were separated and sorted, however were not included in subsequent analyses as they are considered nodule fauna (due to their dependence on the substratum) and not sediment fauna. Serpulid polychaetes were excluded from analyses for the same reason, except for certain species which were determined to be free-living. To minimize systematic error in the sorting process, a subsection (approximately 10%) of sample residues were re-sorted by different experts to ensure no fauna were missed.

Once sorted, individual groups were split between institutions for subsequent identification by taxonomic experts, with annelids and molluscs sent to the University of Gothenburg (Sweden), arthropods to the National Oceanography Centre (Southampton, UK) and all other groups remaining at the NHMUK.

### Taxonomic identification

There are no published field guides to the fauna of the CCZ and approximately 90% of the known species are considered undescribed[11]. As such, 'identification' of the species by traditional methods is not possible (for example, comparison with published literature). Taxonomic assignments therefore considered information drawn from both molecular and morphological investigation of every specimen. Using a morphology-based approach, every specimen was assigned to the lowest operational taxonomic unit (OTU; hereby simply referred to as 'species'), each representing a species. In cases where several specimens were identified as the same species, the voucher code of the best-preserved specimen was used as the identifier of that species, for example: *Ophiohelus* sp. [NHM_8098]. This avoids confusion with the use of sp. A, sp. B, sp. C and so on, where informal and confusing synonyms can easily arise. Open nomenclature was used following ref. 60 and species authorities can be found on WoRMS[61]. At least one representative from each species, and specimens which could not be assigned by morphology to a species, were then DNA barcoded using standard genetic markers (for example: cytochrome c oxidase subunit 1, 16S ribosomal DNA) for different taxa (for example, refs. 62–66). These data have been and are being published in a series of taxonomic publications separate to this study (for example, refs. 63,67,68). The genetic sequences generated were compared against an internal reference database to further guide identifications.

### Data analysis

A macrofauna sensu stricto concept was used when defining the specimens to be included in analyses, this included any metazoan retained on a 300 μm sieve, but excluded traditionally considered meiofaunal taxa such as Ostracoda, Copepoda, Nematoda and Halicarida, as well as known sessile nodule fauna (the subject of a separate study) and pelagic or likely pelagic contaminants (such as Hyperiid amphipods). This approach allows for the greatest comparison with published data. Processing of boxcore samples with regards to sliced layers differs between studies[7–9,16,69] and, as such, the data from all layers (0–10 cm) are summarized together for all analyses. All specimen data used for analyses can be found in Darwin Core (DwC) format in Supplementary Table 8.

All analyses were conducted using R v.4.4.1., using the packages vegan[70], SpadeR[71], iNEXT[72], BiodiversityR[73], indicspecies[74] and rstatix[75].

Following the work of Underwood[28,29], we consider an 'impact' to be defined as a difference in the change of mean abundance/diversity, or time course of mean abundance/diversity, in a defined 'impacted' area (in this instance, the mining track or plume area) from before to after the disturbance, compared with such changes that occur from before to after in the control locations. While the same sites were sampled at several time points, we do not consider them to be part of a 'repeated measures' sample design. Boxcores were randomly taken within the predefined areas and as deep-sea macrofaunal community composition can vary at the centimetre scale[14] we consider samples to be independent. Samples from the PRZ and FFE sites are included in all

figures for visual comparison, however, were excluded from statistical analyses because of a lack of temporal replication.

### Macrofaunal density

To examine differences in total faunal abundance the full dataset was used, including specimens which could only be identified to higher taxonomic levels (for example, genus or family level, such as 'Polychaeta indet.'). Faunal abundances were transformed into densities (individuals per m$^2$) before all calculations.

A two-way ANOVA with site and time as fixed factors was used to assess if there were significant interactions between site and sampling time on macrofaunal abundance. As the plume and track sites were nested within the CTA, the two-way ANOVA was conducted twice, excluding the plume or the track, to examine changes over time within the CTA. Where significant factor effects were found, one-way ANOVAs were conducted on site at each level of time and time at each level of site. These were followed by pairwise $t$-tests to test for significant differences between sites at each time point and between time points for each site. No adjustment of $P$ values was performed, as the argument in favour of adjusting for type I errors applies to random distributions, which is seldom the case when studying living systems[76,77].

Before each analysis, normality and homogeneity of variances were checked with Shapiro–Wilkes and Levene's tests, respectively. Levene's tests were performed on data within each time point and within each site across all time points. Transformation could not stabilize significantly heterogeneous variances in all identified cases and so untransformed data were analysed. Where significant heterogeneity of variance was found, a non-parametric Welches ANOVA was performed in place of the specific one-way ANOVA. For the two-way ANOVA, results were considered robust if not significant (at $P > 0.05$), since the probability of type II error is not affected by heteroscedasticity, or significant at $P < 0.01$ (instead of $P < 0.05$), to compensate for increased probability of type I error[78]. For all other analyses a significance level of 0.05 was used.

As increased variability can be an indicator of disturbance in marine communities[30], pairwise Levene's tests were performed between samples to identify significant differences in variance.

### Biodiversity and species richness

For biodiversity analyses, we examined both species richness (the total number of species, irrespective of abundance) and species diversity, measures including both the number of species and degree of evenness or dominance. A reduced dataset was used for these analyses, including only specimens identified to species level, either a formally described species (for example, *Vesicomya galatheae*) or an identified OTU (for example, Acrocirridae sp. [NHM_559]).

On the basis of abundance data from an assemblage, calculation of species richness (both observed and estimated) is statistically difficult, especially for highly diverse assemblages with many rare species, as is typically the case for the abyssal benthos[10,79]. As such, we examined biodiversity and the way it varies both spatially and temporally in several ways at different scales, considering species richness and diversity, diversity accumulators and sample-size independent measures.

To compare biodiversity between sites and time points in terms of the observed diversity per unit area, species richness ($S$), Gini-Simpson (1-$D$) and Simpson's evenness ($E_{1/D}$) indices were calculated for each boxcore sample. To detect spatial and temporal changes in richness and diversity both before and after test mining, these measures of diversity were analysed using two-way and one-way ANOVAs, pairwise $t$-tests and Levene's tests, as described for abundance data. Indices were also calculated for the communities as a whole, grouping all boxcores into one sample.

Both individual-based and sample-based rarefaction curves were computed for each sampled area and time point, to provide sample-size independent measures of diversity based on accumulation rates. On

the basis of these data, the expected number of species was calculated for 100 individuals (ES(100)) from each area and the expected number of species to be sampled from four boxcores (ES(4bc)). In addition, non-parametric species richness estimators were computed for each area to represent local diversity. While rarefaction compares observed richness among samples, richness estimators evaluate the total richness of a community. Abundance-based estimators included Chao1[80] and Chao1-bc (Chao1 bias-corrected)[81]. Incidence-based estimators included first- and second-order Jackknife[82].

To visualize the dominance structure of species within communities, k-dominance curves were plotted. k-dominance, or cumulative curves, are the cumulative relative abundance of species plotted against the species rank, where the most elevated curve has the lowest diversity[83].

### Community composition

Community composition analyses used the same reduced 'species-level' dataset as used in diversity analyses. Analyses were based on a resemblance matrix using the Bray–Curtis similarity index after square-root transformation of abundance data. This transformation procedure allows for all species, including singletons, to contribute to the similarity matrix while giving the most common species greater weight. Multivariate differences in community structure between sites and time points were determined graphically using non-metric multidimensional scaling (nMDS) and quantified using PERMANOVA[84], with P values obtained by 9,999 permutations of the residuals. PERMANOVA tests were configured as described for univariate analyses, with site and time as fixed factors. Following PERMANOVA analysis, pairwise comparisons between communities were made using pairwise.adonis2, a wrapper function for multilevel pairwise comparison using adonis2 from the package vegan[85]. Significant differences in multivariate dispersion were tested for using the betadisper function in the package vegan, which implements the PERMDISP method of ref. 86, a multivariate analogue of a Levene's test, to analyse multivariate homogeneity of group dispersions. This was followed by a permutation test for significance and pairwise t-tests to identify significant differences in dispersion. Multivariate dispersion has been used previously to detect disturbance responses in deep-sea benthic communities[33].

Indicator species analysis[87] was performed using multilevel pattern analysis[88] to determine if any taxa were significantly associated with any site or time point. The IndVal index measures the strength of association between a taxonomic unit and a site group and is based on the product of the mean abundance and relative frequency of occurrence of each group within a given dataset. Statistical significance of the relationship between taxa and site is tested using Monte Carlo randomizations with 9,999 permutations[87]. Analysis was conducted using abundance data for polychaete families and for taxa identified to Class level (or higher) to improve statistical power, due to the low abundances of each species. The full dataset, including all specimens, was used for this analysis.

### Reporting summary

Further information on research design is available in the Nature Portfolio Reporting Summary linked to this article.

## Data availability

All data generated for this study and used in analyses are available in Supplementary Tables.

## Code availability

Data handling and analyses were implemented using standard methods, software tools and code functions detailed in Methods.

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

## Acknowledgements

We would like to thank the masters, crew and technical staff aboard the *Maersk Launcher* and MV *Island Pride* for their outstanding support, C. Dalgleish and T. Adamson for managing at-sea operations and L. Marsh for managing the entire NORI-D environmental programme. We thank all researchers aboard the vessels who assisted with boxcore processing, sorting and sieving samples. We acknowledge the continued support from the NHMUK Consultancy team (H. Rousham, R. Fryer and J. Herrington). E.C.D.S. was supported through funding from NERC (grant no. NE/S007210/1).

## Author contributions

A.G.G. and T.G.D. conceived and designed the study. A.G.G., T.G.D. and T.H. acquired funding. E.C.D.S., G.B.-C., R.D., M.R. and C.M.B.B. collected and processed samples at sea, led by H.W. L.N., L.D.K., E.C.D.S., M.B.A. and C.M.B.B. processed samples in the laboratory. L.N., T.H., H.W., G.V.D., A.S-S., G.B.-C. and R.D. conducted species identifications. E.C.D.S. performed all analyses and data visualizations. E.C.D.S. led the writing with A.G.G. and all authors reviewed, edited and approved the final version.

## Competing interests

This work was funded by TMC through its subsidiary NORI. NORI holds exploration rights to the NORI-D contract area in the CCZ regulated by the ISA and sponsored by the government of Nauru. It is required by international law to carry out the environmental work to the best standard and to make data and analyses available to all. The funders had no influence on data analysis, interpretation or presentation.

## Additional information

**Extended data** is available for this paper at https://doi.org/10.1038/s41559-025-02911-4.

**Correspondence and requests for materials** should be addressed to Eva C. D. Stewart or Adrian G. Glover.

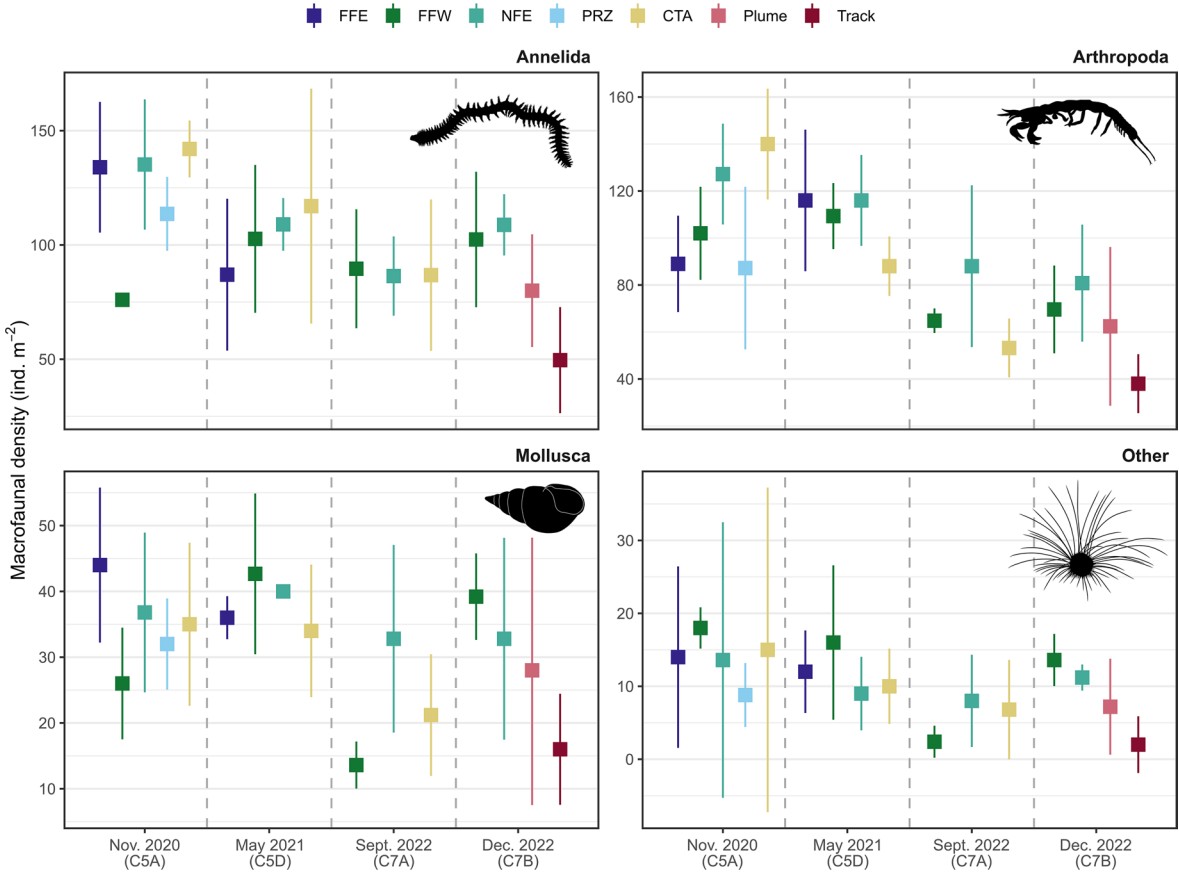

**Extended Data Fig. 1 | Benthic macrofaunal density per m² split by phylum at each sampled site and time point.** Squares indicate mean and bars show standard deviation. 'Other' includes Echinodermata, Nemertea, Bryozoa, Cnidaria, and un-identifiable metazoans. Sample sizes (n): C5A FFE = 4, C5A FFW = 2, C5A NFE = 4, C5A PRZ = 5, C5A CTA = 4, C5D FFE = 4, C5D FFW = 3, C5D NFE = 4, C5D CTA = 4, C7A FFW = 5, C7A NFE = 5, C7A CTA = 10, C7B FFW = 5, C7B NFE = 5, C7B Plume = 5, C7B Track = 10.

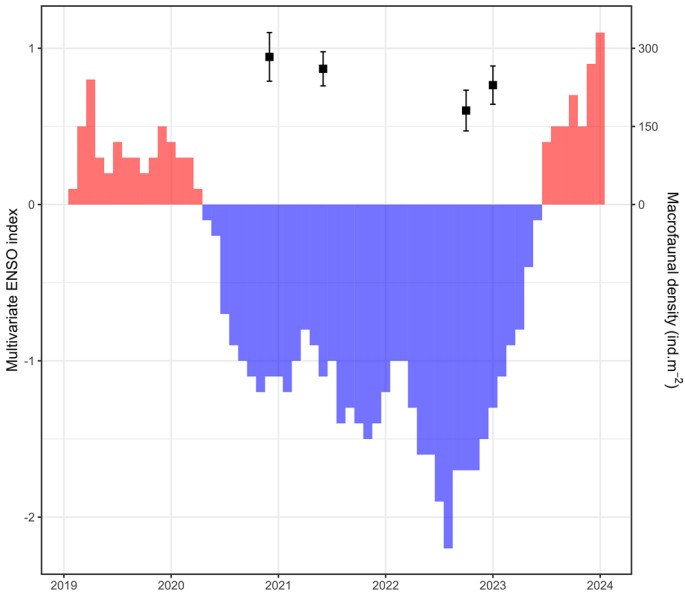

**Extended Data Fig. 2 | Bimonthly multivariate ENSO (El Niño Southern Oscillation) index (MEI) compared with mean macrofaunal densities in NORI-D.** MEI values extracted from https://psl.noaa.gov/enso/mei/. Squares indicate mean and bars show standard deviation of grouped macrofaunal densities from all sampled sites within NORI-D, excluding samples taken from within sites impacted by the Oct. 2022 mining test (Track and Plume). Sample sizes (n) from left to right = 20, 15, 20, 10.

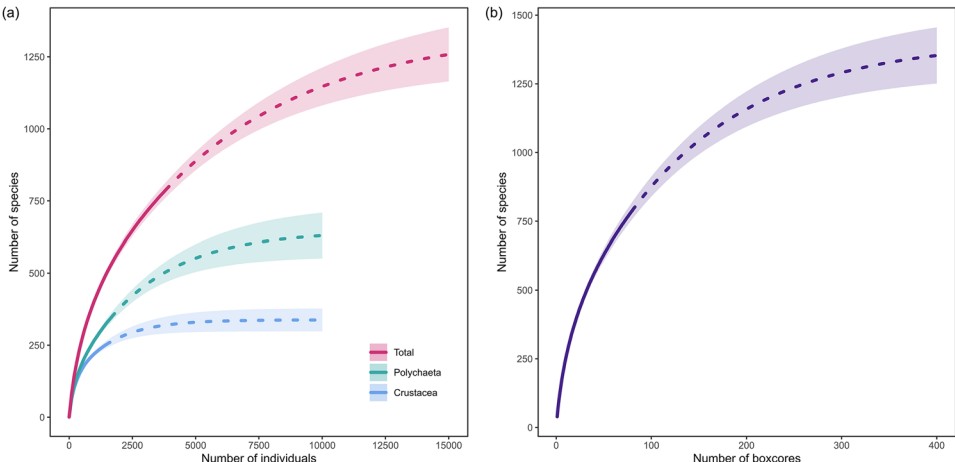

**Extended Data Fig. 3 | Rarefied species diversity across NORI-D.** Rarefaction curves of number of species as a function of (**a**) number of specimens sampled, and (**b**) number of boxcore samples in each area. Solid lines are observed values, dashed lines represent extrapolated values, and shading shows 95% confidence intervals.

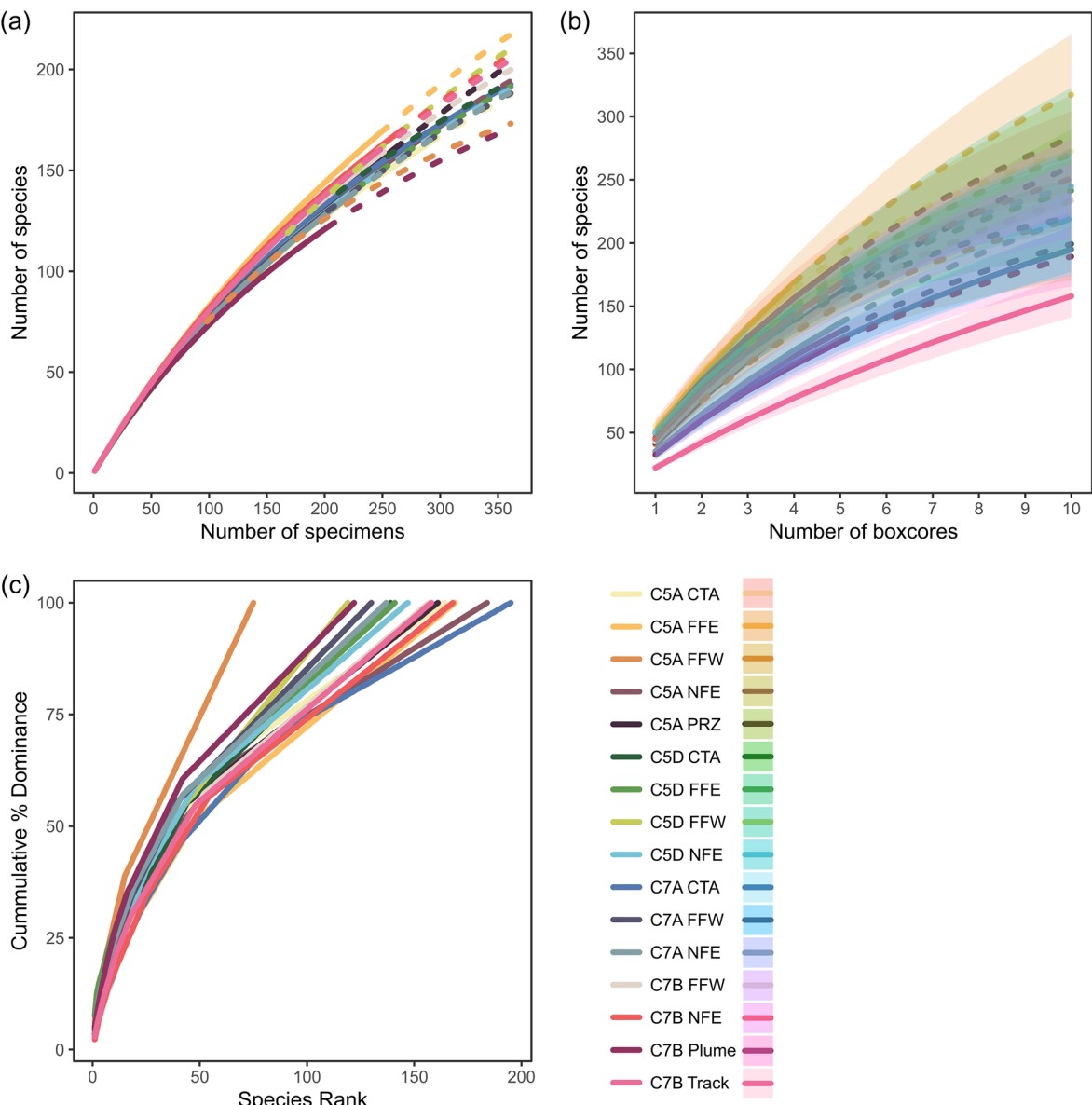

**Extended Data Fig. 4 | Complete rarefaction and *k*-dominance curves for all sites at all time points.** Rarefaction curves of number of species as a function of (**a**) number of specimens sampled, and (**b**) number of boxcore samples in each area. Solid lines are observed values, dashed lines represent extrapolated values, and shading shows 95% confidence intervals. (**c**) *k*-dominance curves.

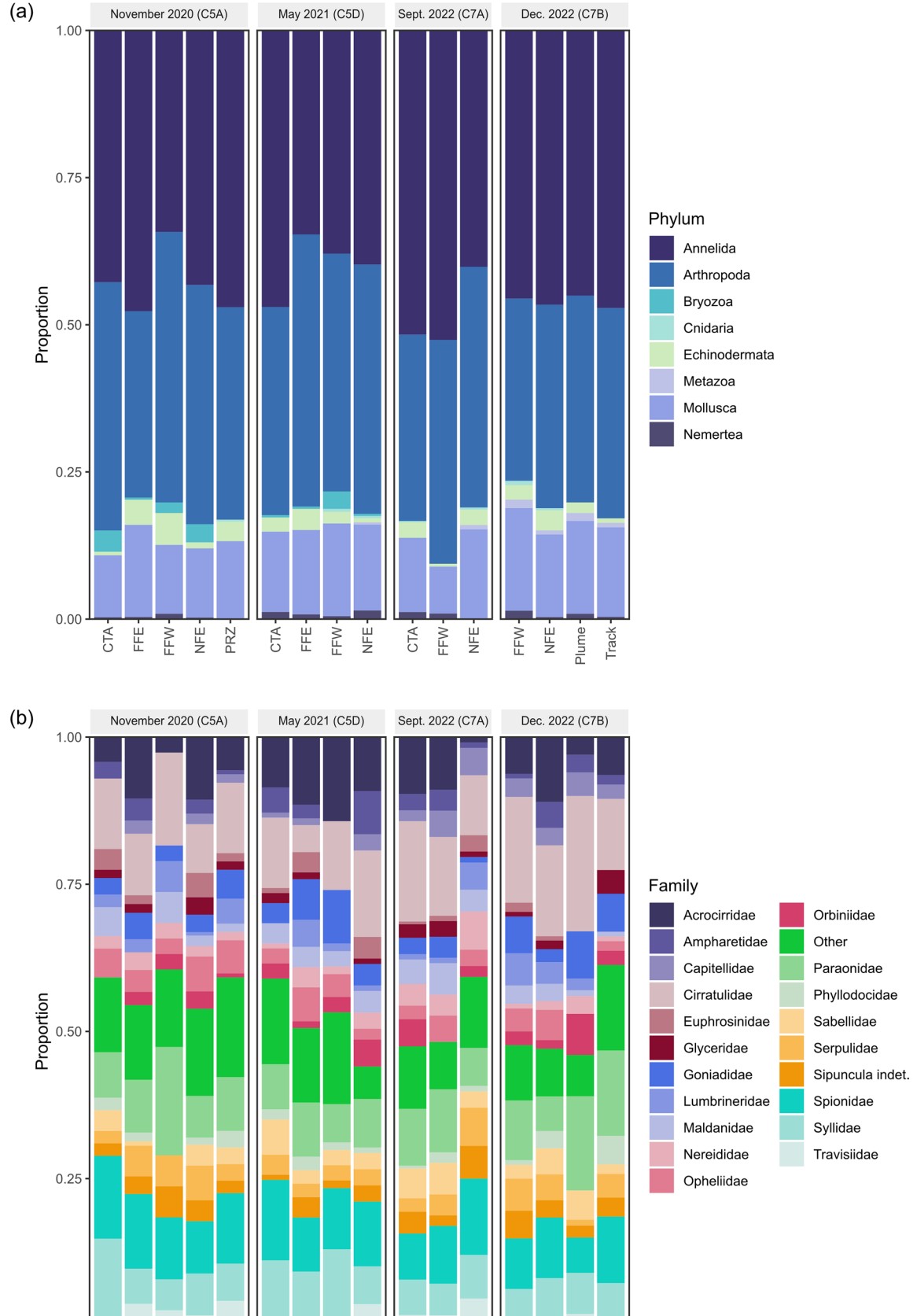

**Extended Data Fig. 5 | Relative proportions of taxonomic groups.** Relative proportion of invertebrate macrofaunal (**a**) phyla, and (**b**) annelid families across all sampled sites and time points.

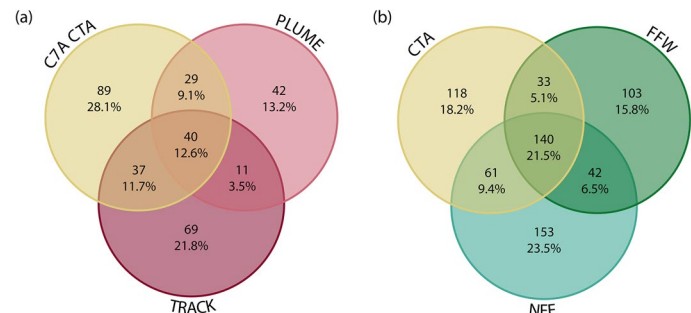

**Extended Data Fig. 6 | Venn diagrams of shared species between sampled sites and times.** Comparisons between (**a**) the CTA pre-impact (September 2022, C7A) and the post-impact track and plume sites; and (**b**) the CTA, FFW, and NFE sites, with all sampling times combined.

# Reporting Summary

## Statistics

For all statistical analyses, confirm that the following items are present in the figure legend, table legend, main text, or Methods section.

| n/a | Confirmed | |
|---|---|---|
| ☐ | ☒ | The exact sample size (*n*) for each experimental group/condition, given as a discrete number and unit of measurement |
| ☐ | ☒ | A statement on whether measurements were taken from distinct samples or whether the same sample was measured repeatedly |
| ☐ | ☒ | The statistical test(s) used AND whether they are one- or two-sided<br>*Only common tests should be described solely by name; describe more complex techniques in the Methods section.* |
| ☒ | ☐ | A description of all covariates tested |
| ☐ | ☒ | A description of any assumptions or corrections, such as tests of normality and adjustment for multiple comparisons |
| ☐ | ☒ | A full description of the statistical parameters including central tendency (e.g. means) or other basic estimates (e.g. regression coefficient) AND variation (e.g. standard deviation) or associated estimates of uncertainty (e.g. confidence intervals) |
| ☐ | ☒ | For null hypothesis testing, the test statistic (e.g. *F*, *t*, *r*) with confidence intervals, effect sizes, degrees of freedom and *P* value noted<br>*Give P values as exact values whenever suitable.* |
| ☒ | ☐ | For Bayesian analysis, information on the choice of priors and Markov chain Monte Carlo settings |
| ☒ | ☐ | For hierarchical and complex designs, identification of the appropriate level for tests and full reporting of outcomes |
| ☒ | ☐ | Estimates of effect sizes (e.g. Cohen's *d*, Pearson's *r*), indicating how they were calculated |

*Our web collection on statistics for biologists contains articles on many of the points above.*

## Software and code

Policy information about availability of computer code

| Data collection | No software was used to collect data. |
|---|---|
| Data analysis | R version 4.4.1<br>R package vegan version 2.6-10<br>R package rstatix version 0.7.2<br>R package SpadeR version 0.1.1<br>R package indicspecies version 1.8<br>R package iNEXT version 3.0.1<br>R package BiodiversityR version 2.17-2 |

For manuscripts utilizing custom algorithms or software that are central to the research but not yet described in published literature, software must be made available to editors and reviewers. We strongly encourage code deposition in a community repository (e.g. GitHub). See the Nature Portfolio guidelines for submitting code & software for further information.

## Data

Policy information about availability of data
All manuscripts must include a data availability statement. This statement should provide the following information, where applicable:

- Accession codes, unique identifiers, or web links for publicly available datasets
- A description of any restrictions on data availability
- For clinical datasets or third party data, please ensure that the statement adheres to our policy

> All data generated for this study and used in analyses are available in the supplementary materials.

## Research involving human participants, their data, or biological material

Policy information about studies with human participants or human data. See also policy information about sex, gender (identity/presentation), and sexual orientation and race, ethnicity and racism.

| | |
|---|---|
| Reporting on sex and gender | *Use the terms sex (biological attribute) and gender (shaped by social and cultural circumstances) carefully in order to avoid confusing both terms. Indicate if findings apply to only one sex or gender; describe whether sex and gender were considered in study design; whether sex and/or gender was determined based on self-reporting or assigned and methods used.*<br>*Provide in the source data disaggregated sex and gender data, where this information has been collected, and if consent has been obtained for sharing of individual-level data; provide overall numbers in this Reporting Summary. Please state if this information has not been collected.*<br>*Report sex- and gender-based analyses where performed, justify reasons for lack of sex- and gender-based analysis.* |
| Reporting on race, ethnicity, or other socially relevant groupings | *Please specify the socially constructed or socially relevant categorization variable(s) used in your manuscript and explain why they were used. Please note that such variables should not be used as proxies for other socially constructed/relevant variables (for example, race or ethnicity should not be used as a proxy for socioeconomic status).*<br>*Provide clear definitions of the relevant terms used, how they were provided (by the participants/respondents, the researchers, or third parties), and the method(s) used to classify people into the different categories (e.g. self-report, census or administrative data, social media data, etc.)*<br>*Please provide details about how you controlled for confounding variables in your analyses.* |
| Population characteristics | *Describe the covariate-relevant population characteristics of the human research participants (e.g. age, genotypic information, past and current diagnosis and treatment categories). If you filled out the behavioural & social sciences study design questions and have nothing to add here, write "See above."* |
| Recruitment | *Describe how participants were recruited. Outline any potential self-selection bias or other biases that may be present and how these are likely to impact results.* |
| Ethics oversight | *Identify the organization(s) that approved the study protocol.* |

Note that full information on the approval of the study protocol must also be provided in the manuscript.

# Field-specific reporting

Please select the one below that is the best fit for your research. If you are not sure, read the appropriate sections before making your selection.

☐ Life sciences  ☐ Behavioural & social sciences  ☒ Ecological, evolutionary & environmental sciences

For a reference copy of the document with all sections, see [nature.com/documents/nr-reporting-summary-flat.pdf](http://nature.com/documents/nr-reporting-summary-flat.pdf)

# Ecological, evolutionary & environmental sciences study design

All studies must disclose on these points even when the disclosure is negative.

| | |
|---|---|
| Study description | This study aims to examine if changes in benthic invertebrate abundance, diversity, and community structure occurred following the test of a commercial prototype polymetallic nodule collector on the abyssal seafloor. To achieve this, seafloor samples were collected using a 50cmx50cm boxcore at 4 sampling times across a two year period. A random stratified sampling design was used, with data collected from four control sites chosen to be spatially equivalent to the area being impacted by the mining disturbance, and one impacted site. A total of 80 quantitative samples were taken, and all individual animals within each sample were identified to a species unit and then used for statistical analysis. |
| Research sample | Benthic invertebrate macrofauna (>300µm) specimens were obtained from 80 boxcore samples of the abyssal seafloor in the Clarion-Clipperton Zone (CCZ), eastern Pacific Ocean. Individuals were identified by taxonomists to species units, totaling 789 species from a total of 4354 specimens. |
| Sampling strategy | Sampling procedures followed standard protocols for sampling abyssal macrofauna in the CCZ, see Glover et al. 2015 (https://doi.org/10.3390/jmse4010002). Sampling strategy was based on International Seabed Authority guidelines for establishing |

| | |
|---|---|
| | appropriate environemental baselines for impact assessments, and established protocols for Before-After Control-Impact (BACI) sampling designs. |
| Data collection | Data were collected on four seagoing expeditions to the CCZ. Samples at sea were collected by ECDS, HW, RD, GBC, CMBB, and MR. Samples were processed in the lab as detailed in the methods by ECDS, HW, CMBB, LK, MBA, GBC, RD, LN, GVD, ASS, and TH. |
| Timing and spatial scale | The study was conducted in the exploration contract area NORI-D, licenced by the International Seabed Authority (ISA) to Nauru Ocean Resources Inc. The NORI-D contract area is located in the south-eastern CCZ, occupying an area of approximately 27000 km2, and with water depth spanning 2959 – 4602 m. Samples were collected across four expeditions to the NORI-D area spanning a total period of two years. Campaign 5A (C5A) took place from October to November 2020 aboard the Maersk Launcher, and campaign 5D (C5D) took place from April to June 2021 also aboard the Maersk Launcher. Campaigns 7A (C7A), and 7B (C7B) both took place on the MV Island Pride, from August to September 2022 and November to December 2022 respectively. Samples were collected from 5 sampling blocks, each approximately 7.75 km square. |
| Data exclusions | Data were excluded based on predetermined criteria established for abyssal macrofauna datasets. A macrofauna sensu stricto concept was used when defining the specimens to be included in analyses, this included any metazoan retained on a 300 μm sieve, but excluded traditionally considered meiofaunal taxa such as Ostracoda, Copepoda, Nematoda, and Halicarida, as well as known sessile nodule fauna (the subject of a separate study) and pelagic or likely pelagic contaminants (such as Hyperiid amphipods). Arthropods and annelids were only counted if the head was present, and echinoderms only if the oral disc was present. Sessile animals such as bryozoans (excluding Ctenostomatida which are free-living within the sediment), brachiopods, crinoids, and sponges were separated and sorted, however were not included in subsequent analyses as they are considered nodule fauna (due to their dependence on the substratum) and not sediment fauna. Serpulid polychaetes were excluded from analyses for the same reason, except for certain species which were determined to be free-living. |
| Reproducibility | The original dataset generated for this study and used to run analyses is provided in the supplementary material. The software and standard code functions are open source and detailed in the methods. |
| Randomization | Samples were taken from randomised locations within each treatment area. |
| Blinding | Blinding was not applicable for this study. |

Did the study involve field work?  ☒ Yes  ☐ No

# Field work, collection and transport

| | |
|---|---|
| Field conditions | Near-seabed oceanographic conditions are similar across the CCZ, with typical temperatures = 1.48°C, oxygen = 151μmol/kg, and salinity = 34.87 g.kg-1. Samples were collected at depths from 4121-4312m. |
| Location | Samples were all collected in the NORI-D contract area in the Clarion-Clipperton Zones, eastern Pacific Ocean. Sites varied in depth from 4121–4312 m |
| Access & import/export | Samples and data were obtained as part of permitted marine scientific research in areas beyond national jurisdiction (the high seas). The area sampled is licenced for deep-sea mineral exploration to Nauru Ocean Resources Inc. (NORI). NORI holds exploration rights to the NORI-D contract area in the Clarion-Clipperton Zone regulated by the International Seabed Authority and sponsored by the government of Nauru. |
| Disturbance | Disturbance caused directly by the study was minimal, with all procedures following standard protocols requiring minimal impact on the seabed. |

# Reporting for specific materials, systems and methods

We require information from authors about some types of materials, experimental systems and methods used in many studies. Here, indicate whether each material, system or method listed is relevant to your study. If you are not sure if a list item applies to your research, read the appropriate section before selecting a response.

| Materials & experimental systems | | Methods | |
|---|---|---|---|
| n/a | Involved in the study | n/a | Involved in the study |
| ☒ | ☐ Antibodies | ☒ | ☐ ChIP-seq |
| ☒ | ☐ Eukaryotic cell lines | ☒ | ☐ Flow cytometry |
| ☒ | ☐ Palaeontology and archaeology | ☒ | ☐ MRI-based neuroimaging |
| ☐ | ☒ Animals and other organisms | | |
| ☒ | ☐ Clinical data | | |
| ☒ | ☐ Dual use research of concern | | |
| ☒ | ☐ Plants | | |

# Animals and other research organisms

Policy information about studies involving animals; ARRIVE guidelines recommended for reporting animal research, and Sex and Gender in Research

| | |
|---|---|
| Laboratory animals | The study did not involve laboratory animals. |
| Wild animals | The study involved the collection of invertebrate animals in the field (no cephalopods). Animals were collected by boxcore and extracted from the sediment. Organisms recovered to the ship are killed by the temperature and pressure changes between the seafloor and the surface. Organisms collected for this study were small (< 2 cm) and included arthropods, cnidaria, annelids, mollusca, bryozoa, porifera, and echinoderms. The species identified are provided in the supplementary material. |
| Reporting on sex | N/A. |
| Field-collected samples | Samples collected from the field were stored in 90% ETOH. |
| Ethics oversight | No ethical approval or guidance was needed. |

Note that full information on the approval of the study protocol must also be provided in the manuscript.

# Plants

| | |
|---|---|
| Seed stocks | *Report on the source of all seed stocks or other plant material used. If applicable, state the seed stock centre and catalogue number. If plant specimens were collected from the field, describe the collection location, date and sampling procedures.* |
| Novel plant genotypes | *Describe the methods by which all novel plant genotypes were produced. This includes those generated by transgenic approaches, gene editing, chemical/radiation-based mutagenesis and hybridization. For transgenic lines, describe the transformation method, the number of independent lines analyzed and the generation upon which experiments were performed. For gene-edited lines, describe the editor used, the endogenous sequence targeted for editing, the targeting guide RNA sequence (if applicable) and how the editor was applied.* |
| Authentication | *Describe any authentication procedures for each seed stock used or novel genotype generated. Describe any experiments used to assess the effect of a mutation and, where applicable, how potential secondary effects (e.g. second site T-DNA insertions, mosiacism, off-target gene editing) were examined.* |

