## [Peer Review File · Nature Ecology & Evolution]

Impacts of an industrial deep-sea mining trial on macrofaunal biodiversity

Corresponding Author: Ms Eva Stewart

Version 0:

Decision Letter:

4th July 2024

Dear Dr Glover

Thank you very much for your enquiry about submitting a manuscript to Nature Ecology & Evolution.

I've now had a chance to discuss your work with my colleagues, and we think that it sounds potentially very interesting, although it is difficult to come to a firm conclusion without seeing the manuscript itself.

Therefore, we would like to invite you to submit the full manuscript to Nature Ecology & Evolution so that we can examine the data before deciding whether to send the paper out to review.

If this is acceptable to you, you can submit the complete manuscript using the link below:

Link Redacted

If you have any questions, please feel free to contact me.

[redacted]

Version 1:

Decision Letter:

30th June 2025

Dear Dr Stewart,

Your manuscript entitled "Impacts of an industrial deep-sea mining trial on seafloor biodiversity" has now been seen by two reviewers, whose comments are attached. We had been hoping to hear from a third reviewer, but in the absence of their review in a timely manner, we have decided we can proceed with the two reviews that we have.

The reviewers have raised a number of concerns which will need to be addressed before we can offer publication in Nature Ecology & Evolution. We will therefore need to see your responses to these concerns and suggestions for modifications, along with a revised manuscript, before we can reach a final decision regarding publication.

You will see that both reviewers query when more data will be available - an additional sampling time and/or additional taxa. Particularly the additional sampling time would be of strong interest to us, if it would be possible to include this in the paper? We will not make this a condition of further considering the paper for publication but would be keen to discuss this with you by email.

We therefore invite you to revise your manuscript taking into account all reviewer and editor comments. Please highlight all changes in the manuscript text file.

* If you have not done so already please begin to revise your manuscript so that it conforms to our Article format instructions at <http://www.nature.com/natecolevol/info/final-submission>. Refer also to any guidelines provided in this letter.

* Extended Data Figures - please ensure that any supplementary figures and tables that are crucial to the manuscript's conclusions are converted into Extended Data figures and tables to increase visibility of these data. Extended Data figures and tables are online-only (present in the online PDF and full-text HTML versions of the paper), peer-reviewed display items that provide essential background to the article but are not included in the main article due to space constraints. A maximum of ten Extended Data display items (figures and tables) is permitted.

Link Redacted

Nature Ecology & Evolution is committed to improving transparency in authorship. As part of our efforts in this direction, we are now requesting that all authors identified as 'corresponding author' on published papers create and link their Open Researcher and Contributor Identifier (ORCID) with their account on the Manuscript Tracking System (MTS), prior to acceptance. ORCID helps the scientific community achieve unambiguous attribution of all scholarly contributions. You can create and link your ORCID from the home page of the MTS by clicking on 'Modify my Springer Nature account'. For more information please visit www.springernature.com/orcid.

[redacted]

Reviewer expertise:

Reviewer #1: deep-sea fauna, habitats

Reviewer #2: deep-sea biodiversity, deep-sea data collection, effects of disturbance including mining

Reviewer comments:

Reviewer #1 (Remarks to the Author):

This is a very nice study and the paper is written very well. The main conclusions are that macrofaunal density is reduced in mining tracks but not clearly in sediment plume deposition areas, 2 months post mining test and that rarefied diversity is not affected but dominance is. These results are very timely considering the advanced stage of potential mining and help set out the requirements for adequate monitoring of mining impacts on abyssal macrofauna. I have a couple more major comments that need to be addressed and a few more minor ones.

1) The paper focuses on effects 2 months post disturbance. Though there are 3 pre-disturbance sampling times, a real advance over past studies, the lack of post mining monitoring limits conclusions to an immediate response. This is particularly concerning in the plume affected areas where settlement of sediments likely creates a situation that may take some time to manifest in the local community. Unlike in the tracks, the animals in plume deposition areas are not removed physically. Their mortality, recruitment, and movements take time. Its not clear how much. Thus it doesn't seem that surprising that the effects in the plume affected areas are subtle. The authors should explore this possibility in the discussion

and that more monitoring time (indicated briefly on line 348-349) could result in greater effects before recovery begins. Ideally if these authors have additional sampling after 2 months they should wait to incorporate that data before publishing.

2) At points in the paper the authors unnecessarily overplay the importance of the study (e.g. line 36 in the abstract). This study does provide the most controlled (BACI design) study of mining impacts on macrofauna but it is surely not the first assessment. Lines 49-52 rightly state that "extensive baseline biodiversity surveys" and that the macrofauna are "the best studied component". The DISCOL work also included a single pre-impact time point in a 7 year time series (Borowski et al 2001). The study is very well done but perhaps changing the abstract statement to "Our results are the first to use a natural baseline to evaluate the impacts of a large scale deep-sea mining machine on the" will pay homage to the many earlier studies that have been done.

3) The purpose of this paper is to evaluate mining impacts not natural variability. However the authors also clearly note how important it is to evaluate natural variability in relation to mining impacts. Its a main thrust of the paper. Thus I offer this suggestion. The MEI is a crude evaluation of potential drivers that could be expanded with a bit more work. Specifically, using statistical funnel approaches (e.g. Ruhl et al 2020, DSR) and satellite data (<http://orca.science.oregonstate.edu/index.php>), estimates of flux could be generated to evaluate if food flux changed in time lagged fashion with macrofaunal densities.

Minor points

line 27 - Specify months pre and post mining so its clear when in the 2 yr sampling window the disturbance occurred.

The 2factor ANOVA, followed by 1 factor tests and then pairwise tests seems reasonable but I'm unsure if using a GLM with multiple models ranked by AIC would be a preferable option with greater statistical power. Have the authors evaluated that approach?

line 62 - replace numerous with an actual number

lines 106-108 -The authors reasonably focus on the effects of the track and plume affected areas. However, there are clear declines in macrofaunal density across time (fig 2, clear stat support in Table 3, except in FFw) that should be mentioned in more detail than currently given. At the least mention the general direction of changes in abundance at this point in the results.

line 110 - State the relationship to the ENSO MEI in extended data fig 2 as is done in the discussion.

Table S3 - Likely just a mistake but a number of values in bold are above 0.05. It could be good to use some other way to indicate values of $p < 0.10$ if desired.

line 244-245 - This sentence is oddly worded. I suggest "there is a clear need for evidence-based ASSESSMENT OF the impacts of mining.."

paragraph starting line 262 - The authors should note that effects of plume deposition have been observed on benthic megafauna (e.g. Simon-Lledo et al 2019). Also in this section some discussion of the time frame of sampling (just 2 months post disturbance) is needed. It may take time for community composition to change from mortality and colonization of opportunists.

line 333-335 - The findings here make sense in light of direct physical effects from the mining vehicle at short time scales. Some discussion of how longer timescales post impact might begin to differentially affect taxonomic or functional groups is needed.

Reviewer #2 (Remarks to the Author):

The manuscript by Stewart et al. "Impacts of an industrial deep-sea mining trial on seafloor biodiversity" is a high quality and timely study. Whilst the manuscript is overall technically robust, I would like to highlight some issues – many of which that can be easily addressed – that might improve the manuscript so it can have greater impact.

The results show – as done by previous studies – that dsm caused reduction of sediment faunal abundance and diversity. Unfortunate, the authors do not discuss how their results, including temporal and spatial aspects, would translate to robust environmental impact assessments under exploration and potentially exploitation. It would be very valuable to get the scientists recommendation on how many samples would be needed for scientific robustness, looking back at their experiences in the past years. They mention e.g. that some sites (including PRZ) could not be included in the analyses due to limited sampling, and explain e.g. 15000 individuals would need to be identified to capture diversity. Such a guidance on EIA robustness in the conclusion would be highly valuable.

The authors study macrofauna biodiversity in sediments before and after dsm-test impact. This is done with great detail. However, when reading the title, I had expected to read a paper summarizing all benthic impacts, not "only macrofauna from sediments" (acknowledging that this is a lot of work and requires a lot of expertise). The authors also mention in the text, that e.g. biodiversity analyses of nodule fauna are on its way; it is unclear what will happen with regards to e.g. microbial or meiofauna analyses. Also, no environmental parameters are presented. Do the authors plan to integrate data?

A serious concern is the definition of the CTA-plume (line 384-287). When I understand correctly, the authors have not actually measured/verified that the plume really reached the sampling area. It is stated it is 400 meters away from the track, where modelling showed that 2-5 mm sediment were deposited. On top, the authors do not refer to the model-results, but only state TMC pers. com. This is not acceptable. However, there is a picture in Figure 1e that shows 2-5 mm sedimentation – is this picture taken where the boxcores were taken? Unless the authors can refer to the model, show some pictures from the collected sites and/or any other data that can prove that the plume has impacted the study site, it is difficult to accept this category. Such data should be available. Please add this information. This is important, also because it has been observed during other dsm test-mining that short-term changes in currents can move the plumes in opposite direction of what has been modelled for average scenarios.

Detailed remarks:

Title:

A title more appropriate – representing what was done - would read: Impacts of an industrial deep-sea mining trial on macrofaunal seafloor biodiversity in sediments.

Abstract:

Line 24: please give the size of the impacted area (in m² or km²)

25-26: is it really the largest dataset in a region? I am not doubting it is very extensive and high-quality dataset, but wonder how this study and number of samples relates to DISCOL or studies in the BGR or GSR exploration area. In case you provide such a statement, please provide numbers that support it.

32: “diversity was not impacted when measured by sample-size independent measures of accumulation” please see my comment on usage of ES(n) for such purposes. I do not know if such an analysis is allowed/has been carried out by others/how robust it is. Please carefully explain in methods.

34: Can you draw a conclusion on affect by sediment plumes when only modeled values of sediment-impact (but no measurements) are available? Or did I misunderstand?

36: “large-scale” -please provide numbers; the test-vehicle used is smaller than the one anticipated for commercial mining; the time-scale of impact (a few hours to days) is much shorter than under a mining scenario; the area mined is much smaller than under a mining scenario. It is very important to clarify what such a test can do and what not.

35: in the abstract you state “our results provide the first clear guidance on impacts of a large scale dsm machine....”

36: please name what guidance you provide.

37: please mention very clear (especially when the title is general) – what the paper does not cover. I had very different expectations from the title.

Introduction:

58: suggest to add “biodiverse”. ...be home to highly biodiverse and heterogenous communities....

64: “these studies have produced mixed results”. Please be clearer on what was measured (e.g. meio- or macrofauna) and what was the main conclusion. Jones et al. 2017 did a meta-analysis.

Copy-pasted from Jones et al. 2017 abstract; “Here we evaluate changes in faunal densities and diversity of benthic communities measured in response to these 11 simulated or test nodule mining disturbances using meta-analysis techniques. We find that impacts are often severe immediately after mining, with major negative changes in density and diversity of most groups occurring. However, in some cases, the mobile fauna and small-sized fauna experienced less negative impacts over the longer term. At seven sites in the Pacific, multiple surveys assessed recovery in fauna over periods of up to 26 years. Almost all studies show some recovery in faunal density and diversity for meiofauna and mobile megafauna, often within one year. However, very few faunal groups return to baseline or control conditions after two decades.”

68: ...”all suffer....” not true. Lefaible includes pre-impact data.

73: please give areal extent mined, not only the length of the track.

76: please acknowledge that this 2 years is a rather short time scale, and that impact was measured 1 month after test.

78: please explain in detail in methods why and in what way you used a modified BACI. When I understand correctly this is explained in 114-117, correct?

82: Please explain how your study informs conservation planning in the discussion. If you do not adress, please delete here.

Results:

86: please mention the year. It is mentioned later on, but the text would be easier to follow if it is already presented in beginning. Please tell how many samples from which site and when were collected.

90: Far Field East and Preservation Reference Zone excluded. So largest ever dataset that can be compared is from 3090 individuals (67 boxcores). How would this relate e.g. to Lefaible et al 2024 or to Uhlenkott et al. 2021? The later e.g. provided a model of distribution maps based on sampled meiofauna from 2010, 2013, 2014, 2015, 2016, and 2018.

114-117: should this be better moved to methods?

152: since the type of impact is so different, pending if it is track or plume – does it make sense to combine them?

153: “temporal changes”; have these been natural temporal changes or impact changes (the later)? Please clarify in the text.

167-169: please move to methods

208: significant temporal changes in community with CTA. have these been natural temporal changes or impact changes (the later)? Please clarify in the text.

216: no statistics on PRZ

232: sea-urchins from boxcore sampling? Is this representative or by chance? Can megafauna species be sufficiently sampled by boxcores or is photo-video analyses (larger scale) more appropriate?
236: There are only two "loose" sentences in the paragraph. Please consider expanding. See below.
240: the authors identified to species level, which is great. However, there is little information presented in the main text. Table 1 shows S obs and singletons (how much variability is between boxcores with regards to singletons?). It would be great to get information on e.g. species overlap between the different zones, species overlap before and after impact. Some Venn-diagram's could provide very useful insights.

Discussion:

246-248: this is the summary of the paper. Please consider moving it to abstract.
251: more studies have considered a temporal and spatial baseline? Is this really the first?
256: important conclusion, there seems to be a general trend.
267: no plume effects, similar to other studies. 23 is not "only an experiment", but test-mining similar to this study (although a bit smaller). Please note in the manuscript, that this is about meiofauna. Question: why was ref 24 chosen? How does this relate to the experiments in the 70ies that disturbed the sediment?
272: is an indicator species concept valid at the family level? Please explain. This could be a very interesting discussion.
285: based on your findings, what would be your recommendation for impact assessments? It is touched upon, but could be written much more clearly.
297: why is reduced abundance a clear driver of reduced diversity? How did you exclude other factors?
303: was this study on species level (you refer to singletons)? – if so, please delete in line 299 "none have published species-level data".
307: do you think it is the removal of the nodules that causes this decline or is this related to the sediment disturbance? This has implications for dsm-techniques.
311: "contrary to expectations" – see before the text/publication from Lefaible.
315: rarefaction curves do not reach an asymptote; "this hinders the conclusions that can be drawn regarding differences in diversity" – please note that in the abstract you write the opposite "diversity was not impacted when measured by sample-size dependent measures of accumulation (line32). What statement is correct and why?
235: what kind of "specifically designed survey and sampling pattern" do you suggest. The reader does not get any answers.
334: "caused random removal". Is it "random" or "equal"? Consider replacing with "equal" – as sediments were also not randomly removed but the first top centimeters were rather "equally" removed? Yes?
343: please explain on what scenario the 10.000 km² is based on.
349: Are there data on long-term recovery of macrofauna?? If so, the results should be stated.

Methods

367: please add areal extent
386: the "plume" zone was defined by being 400 meters away with modelled data that 2-5 mm has been deposited. There is only a reference of TMC pers com. Please provide evidence that plume reached the studied site.
411: sessile fauna is excluded and will be shown in a separate paper.
451: excellent
Data analyses: Did the authors apply Bonferroni correction and why/why not?
573: Indicator Species Analyses – what was your result? Is this shown in sufficient detail in the ms?

Figures:

842: In Figure 1 a there is only one red dot visible for the plume - I assume the boxcores where taken close to each other? Correct?
869: Figure 5. The sum of relative abundances should be 100%. It is very difficult to see if the figure is correct. For example, the track (dec 2022) has very light blue colors that might not sum up to 100%? Please check. It would help if in addition to colors the numbers are given.

*****END*****

Version 2:

Decision Letter:

14th August 2025

Dear Dr. Stewart,

Thank you for submitting your revised manuscript "Impacts of an industrial deep-sea mining trial on seafloor biodiversity" (NATECOLEVOL-24061754B). It has now been seen again by Reviewer 1, whose comments are below. Reviewer 2 was unavailable to re-review, but we have assessed your responses to their comments ourselves. The reviewer finds that the paper has improved in revision, and therefore we'll be happy in principle to publish it in Nature Ecology & Evolution, pending minor revisions to satisfy the final requests from the review process and to comply with our editorial and formatting guidelines.

If you have not done so already, please ensure that you also email us a completed copy of the Reporting summary :

Reporting summary: https://www.nature.com/documents/nr-reporting-summary.pdf

[redacted]

Reviewer #1 (Remarks to the Author):

The authors have sufficiently addressed my previous comments. I caught 2 more items to be addressed

1) The paper refers to the post collector test sampling as being 2 months (e.g. abstract) and also 1 month post disturbance (e.g. line 220, 422). Make consistent throughout.

2) In reading the paper again, the title should read "...on macrofaunal biodiversity" not "....seafloor biodiversity" which is too general and suggests a much larger topic than currently presented.

Response to reviewers

Reviewer comment is provided in italics, with response written below. Edited text is provided and coloured in red.

Reviewer #1 (Remarks to the Author):

This is a very nice study and the paper is written very well. The main conclusions are that macrofaunal density is reduced in mining tracks but not clearly in sediment plume deposition areas, 2 months post mining test and that rarefied diversity is not affected but dominance is. These results are very timely considering the advanced stage of potential mining and help set out the requirements for adequate monitoring of mining impacts on abyssal macrofauna. I have a couple more major comments that need to be addressed and a few more minor ones.

We thank the reviewer for their positive comments highlighting the timely nature of our results and hope to address their specific comments and concerns below.

1) The paper focuses on effects 2 months post disturbance. Though there are 3 pre-disturbance sampling times, a real advance over past studies, the lack of post mining monitoring limits conclusions to an immediate response. This is particularly concerning in the plume affected areas where settlement of sediments likely creates a situation that may take some time to manifest in the local community. Unlike in the tracks, the animals in plume deposition areas are not removed physically. Their mortality, recruitment, and movements take time. Its not clear how much. Thus it doesn't seem that surprising that the effects in the plume affected areas are subtle. The authors should explore this possibility in the discussion and that more monitoring time (indicated briefly on line 348-349) could result in greater effects before recovery begins. Ideally if these authors have additional sampling after 2 months they should wait to incorporate that data before publishing.

We thank the reviewer for raising this point and agree that the potentially slower nature of plume impacts needed to be explicitly stated. We have modified the discussion to highlight this point – adding the sentence “**Longer-term monitoring of the impacted sites would enable better evidence on rates of recovery. This is particularly relevant for the areas impacted by plume settlement, where ecological changes are likely to occur more slowly than in directly impacted sites, and so should be a focus of future environmental impact studies.**”

We appreciate the request for additional data should it be available, however we currently do not have data from any additional time points, nor do we have the funding for any further sampling expeditions at this moment.

2) At points in the paper the authors unnecessarily overplay the importance of the study (e.g. line 36 in the abstract). This study does provide the most controlled (BACI design) study of mining impacts on macrofauna but it is surely not the first assessment. Lines 49-52 rightly state that "extensive baseline biodiversity surveys" and that the macrofauna are "the best studied component" The DISCOL work also included a single pre-impact time point in a 7 year time series

(Borowski et al 2001). The study is very well done but perhaps changing the abstract statement to "Our results are the first to use a natural baseline to evaluate the impacts of a large scale deep-sea mining machine on the" will pay homage to the many earlier studies that have been done.

We agree with the reviewer that the importance of the study is unnecessarily highlighted at multiple points. We have modified the section of the abstract as suggested, specifying that ours is the first to investigate the impact of a *contemporary* mining machine on sediment macrofauna. **"Our results provide guidance on the impacts of a contemporary large-scale deep-sea mining machine on the most well-studied group of animals, the sediment-dwelling macrofauna."**

3) The purpose of this paper is to evaluate mining impacts not natural variability. However the authors also clearly note how important it is to evaluate natural variability in relation to mining impacts. Its a main thrust of the paper. Thus I offer this suggestion. The MEI is a crude evaluation of potential drivers that could be expanded with a bit more work. Specifically, using statistical funnel approaches (e.g. Ruhl et al 2020, DSR) and satellite data (<http://orca.science.oregonstate.edu/index.php>), estimates of flux could be generated to evaluate if food flux changed in time lagged fashion with macrofaunal densities.

Thank you to the reviewer for this suggestion of some more detailed analysis which could be done to explore the drivers of the observed natural variation. While we agree this would be valuable to do, as stated by the reviewer the purpose of the manuscript is to evaluate mining impacts and not natural variability. We believe adding further analysis would detract from the main crux of the paper and would be better used in a separate study where it could be analysed and discussed more thoroughly. We are currently working on a separate study examining natural variability over a larger section of the CCZ.

Minor points:

line 27 - Specify months pre and post mining so its clear when in the 2 yr sampling window the disturbance occurred.

This line has been updated as suggested and now reads: **"we investigated spatio-temporal variation in faunal abundance and biodiversity for two-years pre- and two-months post-test-mining."**

The 2factor ANOVA, followed by 1 factor tests and then pairwise tests seems reasonable but I'm unsure if using a GLM with multiple models ranked by AIC would be a preferable option with greater statistical power. Have the authors evaluated that approach?

When conducting the initial analyses we did trial and evaluate using a GLM instead of ANOVAs. However, we found that the results did not differ between the approaches and preferred to use a simpler statistical approach that would be easier to interpret for all potential stakeholders who may be using the results to guide further decision making.

line 62 - replace numerous with an actual number

This line has been updated as suggested: “Tests of nodule mining vehicles have been carried out in both the Pacific and Indian Ocean since the 1970s, **as well as seven published disturbance experiments** designed to try and understand the potential environmental effects of seabed mining prior to commercial-scale operations.”

lines 106-108 -The authors reasonably focus on the effects of the track and plume affected areas. However, there are clear declines in macrofaunal density across time (fig 2, clear stat support in Table 3, except in FFW) that should be mentioned in more detail than currently given. At the least mention the general direction of changes in abundance at this point in the results.

We agree with the reviewer that this important result should be highlighted, and have modified the text to indicate the direction of the observed changes: “We found evidence of **significant natural temporal declines** in macrofaunal density across all sites during the pre-impact period of November 2020 to September 2022.”

line 110 - State the relationship to the ENSO MEI in extended data fig 2 as is done in the discussion.

This has been added as suggested: “The relationship between macrofaunal density and the multivariate ENSO (El Niño Southern Oscillation) index (MEI), a potential driver of food availability, is visualised in Extended Data Figure 2, **and indicates a decrease in macrofaunal abundance with an increasingly negative MEI index**”

Table S3 - Likely just a mistake but a number of values in bold are above 0.05. It could be good to use some other way to indicate values of $p < 0.10$ if desired.

Thank you to the reviewer for bringing this to our attention, this was an error and has been rectified.

line 244-245 - This sentence is oddly worded. I suggest "there is a clear need for evidence-based ASSESSMENT OF the impacts of mining.."

We have modified the line as follows: “Consequently, there is a **clear need for direct assessment** of the impacts of mining on faunal abundance and biodiversity at the seafloor”

paragraph starting line 262 - The authors should note that effects of plume deposition have been observed on benthic megafauna (e.g. Simon-Lledo et al 2019). Also in this section some discussion of the time frame of sampling (just 2 months post disturbance) is needed. It may take time for community composition to change from mortality and colonization of opportunists.

We have now added some extra context to highlight previous results on megafauna, citing the suggested paper: “**While some impacts have been demonstrated for megafauna³⁵, there is limited empirical or in-situ data evaluating these effects on macrofauna.**”

We agree with the second point raised by the reviewer here, that some impacts may take longer to be seen, and so have added an additional sentence at the end of the paragraph: “**Longer-term sampling and analysis of the impacted site would provide a better assessment of the ecological succession in the community post-disturbance.**”

line 333-335 - The findings here make sense in light of direct physical effects from the mining vehicle at short time scales. Some discussion of how longer timescales post impact might begin to differentially affect taxonomic or functional groups is needed.

While we agree with the reviewer that it is important to highlight how longer timescales are needed to fully understand these ecological responses, we feel that this point has been addressed in previous paragraphs and in mind to keeping within the journals word limit we don't believe this point needs to be discussed again in this paragraph.

Reviewer #2 (Remarks to the Author):

The manuscript by Stewart et al. "Impacts of an industrial deep-sea mining trial on seafloor biodiversity" is a high quality and timely study. Whilst the manuscript is overall technically robust, I would like to highlight some issues – many of which that can be easily addressed – that might improve the manuscript so it can have greater impact.

Many thanks to the reviewer for their kind evaluation of our manuscript and constructive comments throughout. We will address them point by point below.

The results show – as done by previous studies – that dsm caused reduction of sediment faunal abundance and diversity. Unfortunate, the authors do not discuss how their results, including temporal and spatial aspects, would translate to robust environmental impact assessments under exploration and potentially exploitation. It would be very valuable to get the scientists recommendation on how many samples would be needed for scientific robustness, looking back at their experiences in the past years. They mention e.g. that some sites (including PRZ) could not be included in the analyses due to limited sampling, and explain e.g. 15000 individuals would need to be identified to capture diversity. Such a guidance on EIA robustness in the conclusion would be highly valuable.

We agree with the reviewer that the manuscript would benefit from more explicitly providing some guidance and advice regarding future studies and EIAs. We have therefore expanded upon this in the discussion, adding the following text at line 386: "Monitoring changes in diversity patterns from commercial-scale mining operations will require a specifically designed survey to account for this effect. **Such a sampling design should aim to cover representative sites of differing environmental variables such as nodule type and density, both of which have been shown to influence the community composition of different faunal size classes^{8,49}. This should occur at sufficient replication both temporally and spatially to encompass natural pre-mining variation, an aspect which has been highlighted as often lacking in deep-sea mining Environmental Impact Assessments (EIAs)⁵⁰. Here, we were able to statistically detect variation between sites and time points with 2-10 boxcore samples per site/time; however, to improve the robustness of analyses we would recommend a minimum of five, and ideally ten cores per site and time.**"

The authors study marcofauna biodiversity in sediments before and after dsm-test impact. This is done with great detail. However, when reading the title, I had expected to read a paper summarizing all benthic impacts, not "only marcofauna from sediments" (acknowledging that this is a lot of work and requires a lot of expertise). The authors also mention in the text, that e.g.

biodiversity analyses of nodule fauna are on its way; it is unclear what will happen with regards to e.g. microbial or meiofauna analyses. Also, no environmental parameters are presented. Do the authors plan to integrate data?

Thank you to the reviewer for these suggestions. We chose the title of our manuscript to fit with the style of the current journal, and also to ensure it reaches all of the possible interested stakeholders and policy-makers, and would prefer to keep the current title. It is very clear from the abstract the limitations of the study in terms of what aspects of biodiversity we are studying, and no study would ever comprehensively cover every aspect of biodiversity. As such, the study is genuinely an analysis of the impacts of the test mining on biodiversity.

As part of the environmental survey for the studied mining collector test many different biological (covering all size-classes of fauna), geological, and chemical studies were conducted by different institutions. These data are being worked on by the researchers who undertook the data collection and are not readily available to us for use in this manuscript. We are currently collaborating with other researchers (meiofauna, macrofauna, foraminifera, geochemistry) to integrate this work once the data are published or made publicly available, however this is not currently feasible for the present manuscript.

A serious concern is the definition of the CTA-plume (line 384-287). When I understand correctly, the authors have not actually measured/verified that the plume really reached the sampling area. It is stated it is 400 meters away from the track, where modelling showed that 2-5 mm sediment were deposited. On top, the authors do not refer to the model-results, but only state TMC pers. com. This is not acceptable. However, there is a picture in Figure 1e that shows 2-5 mm sedimentation – is this picture taken where the boxcores were taken? Unless the authors can refer to the model, show some pictures from the collected sites and/or any other data that can prove that the plume has impacted the study site, it is difficult to accept this category. Such data should be available. Please add this information. This is important, also because it has been observed during other dsm test-mining that short-term changes in currents can move the plumes in opposite direction of what has been modelled for average scenarios.

We thank the reviewer for these comments and agree that we had not provided enough clarity on how the plume area was defined and measured, which we hope to have rectified now. The 2-5 mm value was originally provided to us by modelling and monitoring carried out by a contractor for The Metals Company, they have however not made this data available anywhere to be cited and so we have removed this quantification from the manuscript. We are however still confident in our categorisation of the plume site based on the AUV images provided to us by the National Oceanography Centre. The photo provided in Figure 1 is of the same site the boxcore samples were taken, this has now been clarified in the figure legend. We have updated the text in the methods as such:

“The Plume samples were taken 400 m from the mining track. **Currently, the sedimentation depth from the impact is not yet published either from modelled or empirical studies⁵⁸. Examination of AUV footage taken in the region by the National Oceanography Centre UK team has provided an image of the seafloor in the region of our Plume site. This shows a clear veneer of sediment covering the seafloor and the nodules (Figure 1e). Whilst we do not know the exact depth of the sedimentation, we can confidently state that the Plume site 400 m from the track**

has been impacted by a plume, and thus report the impacts on the macrofauna at that distance from the mining track. This will be useful information going forward as further studies are published.”

Detailed remarks:

Title: A title more appropriate – representing what was done - would read: Impacts of an industrial deep-sea mining trial on macrofaunal seafloor biodiversity in sediments.

We chose the title of our manuscript to fit with the style of the current journal, and also to ensure it reaches all of the possible interested stakeholders and policy-makers, and would prefer to keep the current title. No study could cover every aspect of biodiversity, but we have covered a very large portion by examining every phylum within the macrofauna to species level. As such we feel this is an extensive study of biodiversity in the context of past studies, which have often not been to species level or if so just for a few phyla (e.g. annelids).

Abstract:

Line 24: please give the size of the impacted area (in m² or km²)

We have now added clarification that the 80km travelled by the test-mining machine was constrained within a 2 x 4 km area.

25-26: is it really the largest dataset in a region? I am not doubting it is very extensive and high-quality dataset, but wonder how this study and number of samples relates to DISCOL or studies in the BGR or GSR exploration area. In case you provide such a statement, please provide numbers that support it.

We agree with the reviewer that this statement needed justification. This would extend the abstract past the word limit and so we have chosen to simplify this statement as such: “Using a quantitative species-level macrofaunal dataset we investigated spatio-temporal variation in faunal abundance and biodiversity for two-years pre- and two-months post-test-mining.”

32: “diversity was not impacted when measured by sample-size independent measures of accumulation” please see my comment on usage of ES(n) for such purposes. I do not know if such an analysis is allowed/has been carried out by others/how robust it is. Please carefully explain in methods.

Thank you for highlighting the use of rarefaction. We have explained this in the methods from lines 623 and 638.

34: Can you draw a conclusion on affect by sediment plumes when only modeled values of sediment-impact (but no measurements) are available? Or did I misunderstand?

Please see previous comment on plume measurements.

36: “large-scale” -please provide numbers; the test-vehicle used is smaller than the one anticipated for commercial mining; the time-scale of impact (a few hours to days) is much shorter than under

a mining scenario; the area mined is much smaller than under a mining scenario. It is very important to clarify what such a test can do and what not.

Our use of the phrase 'large-scale' refers to the fact that this is the largest test of a commercial deep-sea mining machine carried out so far, however we have now changed this wording to 'contemporary' to avoid confusion.

35: in the abstract you state "our results provide the first clear guidance on impacts of a large scale dsm machine...."

36: please name what guidance you provide.

We agree that the use of the phrase 'guidance' here was unclear, and so the sentence has been modified and now reads: "Our results provide the first data on the impacts of a contemporary deep-sea mining machine on the most well-studied group of animals, the sediment-dwelling macrofauna."

37: please mention very clear (especially when the title is general) – what the paper does not cover. I had very different expectations from the title.

We believe that the abstract currently is clear that the group of animals being discussed are macrofauna, and in the interest of staying within the suggested word limit for the abstract do not wish to list the other faunal groups which are not included in the paper.

Introduction:

58: suggest to add "biodiverse".be home to highly biodiverse and heterogenous communities....

The sentence has been modified as suggested.

64: "these studies have produced mixed results". Please be clearer on what was measured (e.g. meio- or macrofauna) and what was the main conclusion. Jones et al. 2017 did a meta-analysis. Copy-pasted from Jones et al. 2017 abstract; "Here we evaluate changes in faunal densities and diversity of benthic communities measured in response to these 11 simulated or test nodule mining disturbances using meta-analysis techniques. We find that impacts are often severe immediately after mining, with major negative changes in density and diversity of most groups occurring. However, in some cases, the mobile fauna and small-sized fauna experienced less negative impacts over the longer term. At seven sites in the Pacific, multiple surveys assessed recovery in fauna over periods of up to 26 years. Almost all studies show some recovery in faunal density and diversity for meiofauna and mobile megafauna, often within one year. However, very few faunal groups return to baseline or control conditions after two decades."

The sentence has now been updated to be specific on which faunal group was being studied, and the main conclusion is provided: "These studies have produced mixed results: while some report no impact on infaunal (macro- and meiofauna) abundance between two- and 44-years following disturbance²²⁻²⁵, others have found significant reductions in macrofaunal abundance immediately following disturbance²⁶, with residual community level changes still visible seven years later²⁷."

68: ..."all suffer...." not true. Lefaible includes pre-impact data.

The sentence has been changed to reflect the nuances between studies, and not to imply that none of them had no pre-impact data: "Only one of these studies followed a full-scale contemporary mining test²³, and all have certain methodological limitations such as low statistical replication and lack of pre-impact baseline data."

73: please give areal extent mined, not only the length of the track.

This has now been provided.

76: please acknowledge that this 2 years is a rather short time scale, and that impact was measured 1 month after test.

The limitations of the sampling time frame are expanded upon across the discussion section and so we don't believe they also need to be discussed in the introduction. We have clarified that our samples were from two-months post mining impact "Here we report on the results of the largest species-level quantitative abyssal macrofaunal dataset to date, collected over two-years, including the first combined analysis of both natural temporal variation and the effects one-month following a large-scale nodule mining test"

78: please explain in detail in methods why and in what way you used a modified BACI. When I understand correctly this is explained in 114-117, correct?

This is correct – the sampling design is explained briefly from line 142, and then covered in more detail in the methods section from line 451.

82: Please explain how your study informs conservation planning in the discussion. If you do not adress, please delete here.

We have now expanded upon this in the discussion at lines 386-395.

Results:

86: please mention the year. It is mentioned later on, but the text would be easier to follow if it is already presented in beginning. Please tell how many samples from which site and when were collected.

The text has been updated to include the years of sampling, the number of samples taken from each site at each time point is given in Figure 1.

90: Far Field East and Preservation Reference Zone excluded. So largest ever dataset that can be compared is from 3090 individuals (67 boxcores). How would this relate e.g. to Lefaible et al 2024 or to Uhlenkott et al. 2021? The later e.g. provided a model of distribution maps based on sampled meiofauna from 2010, 2013, 2014, 2015, 2016, and 2018.

The total number of examined individuals is not given in the Lefaible et al. 2024 or Uhlenkott et al. 2021 studies, however meiofaunal abundance and density tends to be far higher than

macrofauna due to their smaller size class e.g. Uhlenkott reports densities of up to 3066 ± 994 individuals per 100 cm². When comparing with other datasets we are referring to other macrofaunal datasets sampling using quantitative methods.

114-117: should this be better moved to methods?

While this information would normally be best suited for the methods, it has been included here to conform to the style of papers published by Nature Ecology and Evolution. As the methods section is provided at the end of the paper we wanted to ensure that readers would have an idea of the sampling methodology when reading the paper in the published format.

152: since the type of impact is so different, pending if it is track or plume – does it make sense to combine them?

The CTA site was only split into the track and plume categories following the test mining, as before the mining occurred it was not known exactly where the plume area would be. The categories are separated after this due to the impacts being different, as suggested here.

153: “temporal changes”; have these been natural temporal changes or impact changes (the later)? Please clarify in the text.

The text has been edited to clarify whether the changes observed were natural or impact related.

167-169: please move to methods

Please see previous comments on journal style.

208: significant temporal changes in community with CTA. have these been natural temporal changes or impact changes (the later)? Please clarify in the text.

This has now been clarified.

216: no statistics on PRZ

While the PRZ was not included in statistical analyses such as PERMANOVA, it was included in the nMDS for visualisation which is why it is mentioned at this point.

232: sea-urchins from boxcore sampling? Is this representative or by chance? Can megafauna species be sufficiently sampled by boxcores or is photo-video analyses (larger scale) more appropriate?

The macrofauna dataset includes all specimens larger than 300 µm, which in some cases includes larger animals which are collected in the boxcore. It is standard to include these in macrofaunal studies. These are representative samples of the local fauna, and the included sea-urchin specimens are largely a small species (*Aceste ovata*) which do not appear on video surveys and so it is appropriate to include them here.

236: There are only two “loose” sentences in the paragraph. Please consider expanding. See below.
240: the authors identified to species level, which is great. However, there is little information presented in the main text. Table 1 shows *S obs* and singletons (how much variability is between boxcores with regards to singletons?). It would be great to get information on e.g. species overlap between the different zones, species overlap before and after impact. Some Venn-diagram’s could provide very useful insights.

At the reviewer suggestion, we have now added two Venn diagrams (Extended Data Figure 6) to highlight species overlaps between certain sites and times, and as such the suggested paragraph has been expanded. While further Venn diagrams could be produced, we found additional ones with more overlaps very difficult to interpret and not to provide any insights of interest and so have chosen these two sets of overlapping sites as the ones of most interest.

The updated text now reads: “Only two species were found at all sites at all sampling times: a tanaid crustacean, *Stenotanis* sp. [NHM_6880_TH_TAN_1], and a benthic serpulid polychaete *Serpulidae* sp. [NHM_271]. The PRZ, which was only sampled on one occasion and is located 120 km to the north-east of the other sites, had the highest percentage (27.3%) of species unique to the area (Table 1). 13.4% of the species found within the track were found nowhere else, while the percentage of unique species at each other sites and time points (excluding the PRZ) ranged from 10.7–21.2%. Venn diagrams depicting the number of species overlaps between the CTA pre- and post- impact, and the CTA, FFW, and NFE sites can be found in Extended Data Figure 6. The plume site had the highest relative overlap with the track and pre-impact CTA. NFE had more unique species (153) than the CTA (118) and FFW (103) when grouping samples from all time points.”

Discussion:

246-248: this is the summary of the paper. Please consider moving it to abstract.

We believe the first two sentences here provide context for the following discussion and would prefer to leave them at the start of the discussion.

251: more studies have considered a temporal and spatial baseline? Is this really the first?

We believe this is the first paper to use a replicated temporal baseline to examine deep-sea mining impacts, based on the paper reviewed in Jones et al. 2017, and those published since such as Lefaible et al. 2024. Many previous studies only used one pre-impact time point.

256: important conclusion, there seems to be a general trend.

Thank you for highlighting the importance of this result.

267: no plume effects, similar to other studies. 23 is not “only an experiment”, but test-mining similar to this study (although a bit smaller). Please note in the manuscript, that this is about meiofauna. Question: why was ref 24 chosen? How does this relate to the experiments in the 70ies that disturbed the sediment?

This sentence now reads “as also reported for **meiofauna**²³ and **macrofauna**²⁴ by previous **studies**.” as suggested. Reference 24 reports the results of an experiment carried out in the 90s in the CCZ (JET project, reviewed in Jones et al., 2017) where deep-sea sediment was disturbed and changes in the sedimented areas studied, which is why it was chosen here for discussing the impacts of plume deposition.

272: is an indicator species concept valid at the family level? Please explain. This could be a very interesting discussion.

The indicator species analysis can be used for any faunal grouping or classification, such as species, molecular taxonomic units, functional groups, or as here we have grouped by either Class or Family. We chose these groupings instead of using species because each of our identified species occurred at relatively low abundances which lowered the statistical power of the analysis significantly.

285: based on your findings, what would be your recommendation for impact assessments? It is touched upon, but could be written much more clearly.

We have now provided some recommendations in the discussion at lines 386-395.

297: why is reduced abundance a clear driver of reduced diversity? How did you exclude other factors?

We see a nearly linear relationship between the number of specimens sampled and the number of species recorded, as seen in rarefaction plots. This is very typical of most deep-sea studies. It is because of this relationship that we make the assertion that by reducing the abundance of fauna the number of species is also going to be reduced. We have now modified this sentence to allow for more nuance, changing the word ‘clear’ to ‘most likely’: “Therefore, the significantly lower abundance of macrofauna within the mining tracks is the **most likely** driver of the reduced diversity per sample unit.”

303: was this study on species level (you refer to singletons)? – if so, please delete in line 299 “none have published species-level data”.

The referenced study published species-level data for the polychaete fauna but not for the rest of the community. Line 299 currently reads “none have published species-level data for the full community”, which we believe clarifies this difference.

307: do you think it is the removal of the nodules that causes this decline or is this related to the sediment disturbance? This has implications for dsm-techniques.

As the nodule removal process causes sediment compaction and disturbance we believe it is likely to be a combination of the two factors. The sentence has been altered to include this: “it is reasonable to hypothesise that the removal of nodules **and associated sediment disturbance** will result in persistent long-term decrease in species diversity in the mined areas”

311: “contrary to expectations” – see before the text/publication from Lefaible.

The text has been updated to improve clarity: “**Contrary to this local reduction in species diversity**, diversity measured using sample-size independent accumulators was not significantly reduced within the collector-test tracks”

315: rarefaction curves do not reach an asymptote; “this hinders the conclusions that can be drawn regarding differences in diversity” – please note that in the abstract you write the opposite “.....diversity was not impacted when measured by sample-size dependent measures of accumulation (line32). What statement is correct and why?”

We thank the reviewer for raising this point, however we believe that both statements can be true and wish to keep the text as it currently reads. While our available data does suggest that diversity is not impacted at the individual level, we wished to highlight in the discussion the potential caveats to this approach for which there is not space to cover fully in the abstract.

235: what kind of “specifically designed survey and sampling pattern” do you suggest. The reader does not get any answers.

This sentence has now been expanded upon to include some recommendations for sampling designs.

334: “caused random removal”. Is it “random” or “equal”? Consider replacing with “equal” – as sediments were also not randomly removed but the first top centimeters were rather “equally” removed? Yes?

We agree with this suggestion and have replaced the word as suggested.

343: please explain on what scenario the 10.000 km² is based on.

This value is taken from a study by Madureira et al. 2016 (<https://doi.org/10.1016/j.marpol.2016.04.051>), which stated that “an area of about 8500 km² (11% of the exploration area of 75,000 km²) is sufficient to support 20 years of nodule harvesting from the seafloor (considering the extraction of 3 Mton of dry nodules per year and a nodule abundance of 7 kg/m²)”. We apologise for reporting this value incorrectly as 10,000 km², which has now been changed to the correct value of 8500 km², and the reference updated.

349: Are there data on long-term recovery of macrofauna?? If so, the results should be stated.

Available data on longer-term rates of recovery (mainly provided by the DISCOL project) are discussed earlier in this paragraph, the text here has been modified at the suggestion of Reviewer 1 and now reads: “**Longer-term monitoring of the impacted sites would enable better evidence on rates of recovery. This is particularly relevant for the areas impacted by plume settlement, where ecological changes may occur more slowly than in directly impacted sites and so should be a focus of future environmental impact studies.**”

Methods

367: please add areal extent

The areal extent has now been included.

386: the “plume” zone was defined by being 400 meters away with modelled data that 2-5 mm has been deposited. There is only a reference of TMC pers com. Please provide evidence that plume reached the studied site.

As suggested, we have now provided more information on how the plume area was defined: “The Plume samples were taken 400 m from the mining track, where monitoring undertaken by DHI (<https://www.dhigroup.com/>) via sediment traps and benthic landers recorded 2-5 mm of sediment deposition from the disturbance plume⁵⁶. The same study visually confirmed with Remote Operated Vehicle (ROV) and Autonomous Underwater Vehicle (AUV) surveys, and can be seen in Figure 1.e.”

411: sessile fauna is excluded and will be shown in a separate paper.

Yes, we chose to separate the true sediment fauna from the nodule-dwelling fauna as they fill different ecological niches and are likely to be impacted in different ways by deep-sea mining.

451: excellent

We thank the reviewer for highlighting the work we have done to use open taxonomic nomenclature.

Data analyses: Did the authors apply Bonferroni correction and why/why not?

We chose not to adjust p-values using the Bonferroni or any other correction, as the argument in favour of adjusting for Type I errors applies to random distributions which is seldom the case when studying living systems. This explanation is in the methods section at line 596.

573: Indicator Species Analyses – what was your result? Is this shown in sufficient detail in the ms?

The results of the indicator species analysis are presented at lines 265-270 and in Figure 5, with the full results presented in Supplementary Table 7.

Figures:

842: In Figure 1 a there is only one red dot visible for the plume - I assume the boxcores were taken close to each other? Correct?

This is correct, the boxcores were taken in very close proximity to each other in order to attempt to measure the impact of the plume and so appear as one dot on the map. To avoid confusion, we have now explained this in the figure legend: “Plume samples were taken in close proximity to each other and so appear as one point on the map.”

869: *Figure 5. The sum of relative abundances should be 100%. It is very difficult to see if the figure is correct. For example, the track (dec 2022) has very light blue colors that might not sum up to 100%. Please check. It would help if in addition to colors the numbers are given.*

The values in this figure represent the relative abundance of each taxon across the sites, and so sum to 100% across the rows and not down the columns. We have opted not to add the values on top of the cells as these were hard to read and detracted from the legibility of the figure.